# Principled Probabilistic Imaging using Diffusion Models as Plug-and-Play Priors

**Zihui Wu**[1]  **Yu Sun**[4]  **Yifan Chen**[5]  **Bingliang Zhang**[1]  **Yisong Yue**[1]  **Katherine L. Bouman**[1,2,3]

[1]Department of Computing and Mathematical Sciences, Caltech
[2]Department of Electrical Engineering, Caltech
[3]Department of Astronomy, Caltech
[4]Department of Electrical and Computer Engineering, Johns Hopkins University
[5]Courant Institute of Mathematical Sciences, New York University

## Abstract

Diffusion models (DMs) have recently shown outstanding capabilities in modeling complex image distributions, making them expressive image priors for solving Bayesian inverse problems. However, most existing DM-based methods rely on approximations in the generative process to be generic to different inverse problems, leading to inaccurate sample distributions that deviate from the target posterior defined within the Bayesian framework. To harness the generative power of DMs while avoiding such approximations, we propose a Markov chain Monte Carlo algorithm that performs posterior sampling for general inverse problems by reducing it to sampling the posterior of a Gaussian denoising problem. Crucially, we leverage a general DM formulation as a unified interface that allows for rigorously solving the denoising problem with a range of state-of-the-art DMs. We demonstrate the effectiveness of the proposed method on six inverse problems (three linear and three nonlinear), including a real-world black hole imaging problem. Experimental results indicate that our proposed method offers more accurate reconstructions and posterior estimation compared to existing DM-based imaging inverse methods.

## 1  Introduction

Inverse problems arise in many computational imaging applications, where the goal is to recover an image $\boldsymbol{x} \in \mathbb{R}^n$ from a set of sparse and noisy measurements $\boldsymbol{y} \in \mathbb{R}^m$. The relationship between $\boldsymbol{x}$ and $\boldsymbol{y}$ can be described by

$$\boldsymbol{y} = \mathcal{A}(\boldsymbol{x}) + \boldsymbol{n}, \tag{1}$$

where $\mathcal{A}(\cdot) : \mathbb{R}^n \to \mathbb{R}^m$ is the forward operator (linear or nonlinear) and $\boldsymbol{n}$ is the random measurement noise in $\mathbb{R}^m$. Since the sparsity and noisiness of $\boldsymbol{y}$ often lead to significant uncertainty in $\boldsymbol{x}$, it is preferable to sample the posterior distribution $p(\boldsymbol{x}|\boldsymbol{y})$ over all possible solutions based on some prior distribution $p(\boldsymbol{x})$, rather than finding a single deterministic solution. Traditional posterior sampling methods often rely on simple image priors that do not reflect the sophistication of real-world image distributions. On the other hand, diffusion models (DMs) have recently emerged as a powerful tool for modeling highly complex image distributions [33, 63]. Nevertheless, it remains a challenge to turn DMs into reliable imaging inverse solvers, which motivates us to develop a principled Bayesian method that leverages DMs as priors for posterior sampling.

Diffusion models generate samples from a distribution by reversing a diffusion process from the target distribution to a simple (usually Gaussian) distribution [33, 63]. In particular, it estimates a clean image $\boldsymbol{x}_0$ from a noise image $\boldsymbol{x}_T$ by successively denoising noisy images, where $\boldsymbol{x}_t \sim p_t$ is the intermediate noisy image at time $t \in [0, T]$. Reversing diffusion requires one to estimate the time-varying gradient log density (score function) $\nabla \log p_t(\boldsymbol{x}_t)$ along the diffusion process, or $\nabla \log p_t(\boldsymbol{x}_t|\boldsymbol{y})$ in the case of sampling the posterior $p(\boldsymbol{x}|\boldsymbol{y})$.

38th Conference on Neural Information Processing Systems (NeurIPS 2024).

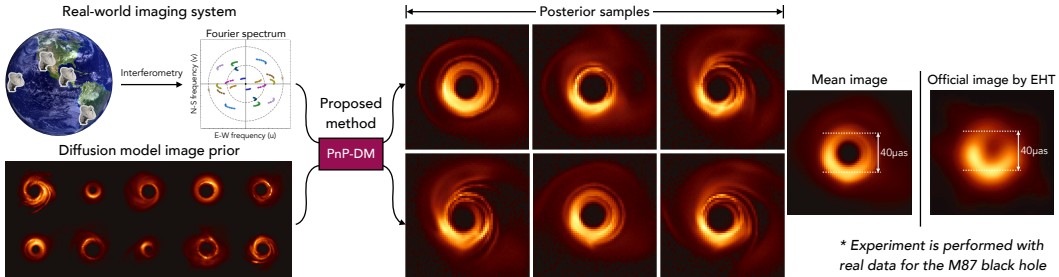

Figure 1: Demonstration of the proposed method, PnP-DM, for posterior sampling using the real data for the M87 black hole from April 6th, 2017 [21]. The black hole imaging problem is non-convex and highly ill-posed due to severe noise corruption and measurement sparsity. Our method rigorously integrates measurements from a real-world imaging system with an expressive image prior in the form of a diffusion model, which was trained with images from the GRMHD black hole simulation [22] in this case. Besides having high visual quality, our posterior samples accurately capture key features of the M87 black hole such as the bright spot location and ring diameter.

To design generic DM-based inverse problem solvers, most existing methods attempt to approximate the time-varying gradient log density $\nabla \log p_t(\boldsymbol{x_t}|\boldsymbol{y})$ [17, 73, 84, 62, 39, 58, 60, 44, 15, 77, 18, 55, 8]. In particular they first apply Bayes' rule to separate the forward operator from an unconditional prior over the intermediate noisy image $\boldsymbol{x_t}$:

$$\nabla \log p_t(\boldsymbol{x_t}|\boldsymbol{y}) = \nabla \log p_t(\boldsymbol{y}|\boldsymbol{x_t}) + \nabla \log p_t(\boldsymbol{x_t}). \tag{2}$$

By instead aiming to evaluate the right hand side, one can leverage the existing pre-trained DMs for the unconditional term $\nabla \log p_t(\boldsymbol{x_t})$. However, the main challenge in this case is that $\nabla \log p_t(\boldsymbol{y}|\boldsymbol{x_t})$ is intractable to compute in general, as $p_t(\boldsymbol{y}|\boldsymbol{x_t})$ involves an integral over all possible $\boldsymbol{x_0}$'s that could give rise to $\boldsymbol{x_t}$ [17]. Various methods have been proposed to circumvent the intractability and can mostly be categorized into two groups. One group of methods explicitly approximate $\nabla \log p_t(\boldsymbol{y}|\boldsymbol{x_t})$ by making simplifying assumptions [62, 17, 60, 8]. However, even for arguably the finest approximation to date proposed in the recent work [8], it is exact only when the prior distribution $p(\boldsymbol{x})$ is Gaussian. For general prior distributions beyond Gaussian, these methods do not sample the true posterior $p(\boldsymbol{x}|\boldsymbol{y})$. The other group of methods do not make explicit approximations but instead substitute $\nabla \log p_t(\boldsymbol{y}|\boldsymbol{x_t})$ with empirically designed updates where $\boldsymbol{y}$ is treated as a guidance signal [73, 84, 39, 58, 44, 15, 77, 18, 55]. Although these methods may have strong empirical performance, they have deviated from the Bayesian formulation and no longer aim to sample the target posterior. In summary, these existing DM-based inverse methods should be best viewed as *guidance methods*, where the generative process is *guided* towards the regions where the measurement $\boldsymbol{y}$ is more likely to be observed, not as posterior sampling methods [8]. We also note that some recent work considered combing DMs with Sequential Monte Carlo to ensure asymptotic consistency in posterior sampling [11, 23], but the investigation has been limited to linear imaging inverse problems.

**Our contributions**  In this work, we pursue a different path towards posterior sampling with DM priors by proposing a new Markov chain Monte Carlo (MCMC) algorithm, which we call *Plug-and-Play Diffusion Models* (PnP-DM). It incorporates DMs in a principled way and circumvents the approximation required when taking the approach in (2). The proposed algorithm is based on the Split Gibbs Sampler [71] that alternates between two sampling steps that separately involve the likelihood and prior. While the likelihood step can be tackled with traditional sampling techniques, the prior step involves a Bayesian denoising problem that requires careful design. Importantly, we identify a connection between the Bayesian denoising problem and the unconditional image generation problem under a general formulation of DMs presented in [37] (which is referred to as the EDM formulation hereafter). This connection allows us to perform rigorous posterior sampling for denoising using DMs without approximating the generative process and enables the use of a wide range of pretrained DMs through the unified EDM formulation. We present an analysis on the non-asymptotic behavior of PnP-DM by establishing a stationarity guarantee in terms of the average Fisher information. We further demonstrate the strong empirical performance of PnP-DM by investigating three linear and three nonlinear noisy inverse problems, including a black hole interferometric imaging problem involving real data that is both nonlinear and severely ill-posed (see Figure 1). Overall, PnP-DM outperforms existing baseline methods, achieving higher accuracy in posterior estimation.

## 2 Preliminaries

*Split Gibbs Sampler (SGS)* is an MCMC approach developed for Bayesian inference [71]. It is also related to the *Proximal Sampler* [42, 14, 25, 79] and serves as the backbone for the *Generative Plug-and-Play (GPnP)* [6] and *Diffusion Plug-and-Play (DPnP)* [78] frameworks in computational imaging. The goal of SGS is to sample the posterior distribution

$$p(\boldsymbol{x}|\boldsymbol{y}) \propto p(\boldsymbol{y}|\boldsymbol{x})p(\boldsymbol{x}) = \exp(-f(\boldsymbol{x};\boldsymbol{y}) - g(\boldsymbol{x})) \tag{3}$$

where $f(\boldsymbol{x};\boldsymbol{y}) := -\log p(\boldsymbol{y}|\boldsymbol{x})$ and $g(\boldsymbol{x}) := -\log p(\boldsymbol{x})$ are the potential functions of the likelihood and prior distribution, respectively. The dual dependence of (3) on both the likelihood and prior makes it nontrivial to directly sample from it in general. Instead, SGS leverages the composite structure of the posterior distribution by adopting a variable-splitting strategy and considers sampling an alternative distribution

$$\pi(\boldsymbol{x}, \boldsymbol{z}) \propto \exp\left(-f(\boldsymbol{z};\boldsymbol{y}) - g(\boldsymbol{x}) - \frac{1}{2\rho^2}\|\boldsymbol{x} - \boldsymbol{z}\|_2^2\right) \tag{4}$$

where $\boldsymbol{z} \in \mathbb{R}^n$ is an augmented variable and $\rho > 0$ is a hyperparameter that controls the strength of the coupling between $\boldsymbol{x}$ and $\boldsymbol{z}$. We denote the $\boldsymbol{x}$- and $\boldsymbol{z}$-marginal distributions of (4) as $\pi^X(\boldsymbol{x}) := \int \pi(\boldsymbol{x}, \boldsymbol{z})\mathrm{d}\boldsymbol{z}$ and $\pi^Z(\boldsymbol{x}) := \int \pi(\boldsymbol{x}, \boldsymbol{z})\mathrm{d}\boldsymbol{x}$, respectively. As $\rho \to 0$, $\pi^X$ converges to the target posterior $p(\boldsymbol{x}|\boldsymbol{y})$ in terms of total variation distance [71], so one can obtain approximate samples from the target posterior by sampling (4) instead.

SGS samples (4) via Gibbs sampling. Specifically, SGS starts from an initialization $\boldsymbol{x}^{(0)}$ and, for iteration $k = 0, \cdots, K - 1$, alternates between

1. **Likelihood step:** sample $\boldsymbol{z}^{(k)} \sim \pi^{Z|X=\boldsymbol{x}^{(k)}}(\boldsymbol{z}) \propto \exp\left(-f(\boldsymbol{z};\boldsymbol{y}) - \frac{1}{2\rho^2}\|\boldsymbol{x}^{(k)} - \boldsymbol{z}\|_2^2\right)$
2. **Prior step:** sample $\boldsymbol{x}^{(k+1)} \sim \pi^{X|Z=\boldsymbol{z}^{(k)}}(\boldsymbol{x}) \propto \exp\left(-g(\boldsymbol{x}) - \frac{1}{2\rho^2}\|\boldsymbol{x} - \boldsymbol{z}^{(k)}\|_2^2\right)$.

Note that the two conditional distributions separately involve $f(\cdot;\boldsymbol{y})$ and $g(\cdot)$. The likelihood and prior are decoupled so that these two steps can be designed in a modular way. A similar variable-splitting strategy is also adopted in optimization methods such as the Half-Quadratic Splitting (HQS) method [31] and the Alternating Direction Method of Multipliers (ADMM) [30, 7]. In fact, SGS can be viewed as a sampling analogue of HQS. SGS is a principled approach to posterior sampling if the two sampling steps are rigorously implemented.

**Existing works related to SGS** Several works have designed algorithms for solving imaging inverse problems based on SGS [53, 19, 6, 27, 78]. The key distinction among these methods lies in their approaches to the prior step. For instance, the works [53, 6, 27] applied Langevin-based updates for sampling $\pi^{X|Z=\boldsymbol{z}}$ such that the prior information is encoded by either traditional regularizers or off-the-shelf image denoisers. The work [19] tackled the prior step by heuristically customizing a diffusion model (i.e. DDPM [33]) for sampling $\pi^{X|Z=\boldsymbol{z}}$. A concurrent work [78] improved the implementation by devising two diffusion processes that rigorously solve the prior step. Our method differs from [78] by connecting the prior step to the EDM formulation [37]. This connection allows us to seamlessly integrate state-of-the-art DMs as expressive image priors for Bayesian inference through a unified interface, eliminating the need for additional customization for each model and leading to better empirical performance. We also note the recent work [43] that adopted the optimization-based variable-splitting formulation of HQS and utilized general DMs as image priors. We instead considers the SGS formulation from a Bayesian posterior sampling standpoint. Additionally, while SGS-based methods theoretically accommodate general inverse problems, empirical evidence on real-world nonlinear inverse problems remains scarce in the literature. In this work, we demonstrate our method on three nonlinear inverse problems, including a black hole imaging problem. For a more comprehensive review of related works, see Appendix E.

## 3 Method

A schematic diagram for the proposed method is shown in Figure 2. Our method, dubbed PnP-DM, builds upon the SGS framework with rigorous implementations of the two sampling steps and an annealing schedule for the coupling parameter $\rho$. We start with our implementations of the first step for solving both linear and nonlinear inverse problems.

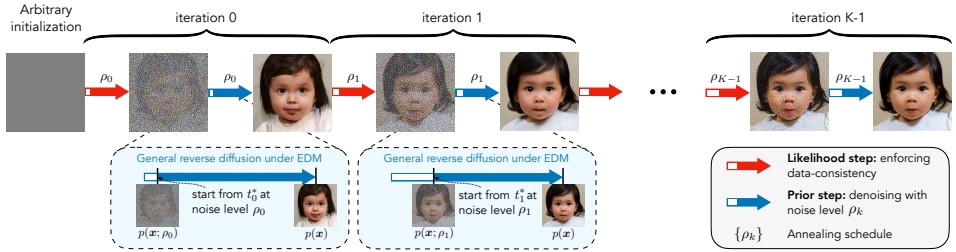

Figure 2: A schematic diagram of our method. Our method alternates between a likelihood step that enforces data consistency and a prior step that solves a denoising posterior sampling problem by leveraging the Split Gibbs Sampler [71]. An annealing schedule controls the strength of the two steps at each iteration to facilitate efficient and accurate sampling. A crucial part of our design is the prior step, where we identify a key connection to a general diffusion model framework called the EDM [37]. This connection allows us to easily incorporate a family of state-of-the-art diffusion models as priors to conduct posterior sampling in a principled way without additional training. Our method demonstrates strong performance on a variety of linear and nonlinear inverse problems.

### 3.1 Likelihood step: enforcing data consistency

For the likelihood step at iteration $k$, we sample

$$\boldsymbol{z}^{(k)} \sim \pi^{Z|X=\boldsymbol{x}^{(k)}}(\boldsymbol{z}) \propto \exp\left(-f(\boldsymbol{z};\boldsymbol{y}) - \frac{1}{2\rho^2}\|\boldsymbol{x}^{(k)} - \boldsymbol{z}\|_2^2\right). \tag{5}$$

**Linear forward model and Gaussian noise**  We first consider a simple yet common case where the forward model $\mathcal{A}$ is linear and the noise distribution is zero-mean Gaussian, i.e. $\mathcal{A} := \boldsymbol{A} \in \mathbb{R}^{m \times n}$ and $\boldsymbol{n} \sim \mathcal{N}(\boldsymbol{0}, \boldsymbol{\Sigma})$. In this case, the potential function of the likelihood term is $f(\boldsymbol{x};\boldsymbol{y}) = \frac{1}{2}\|\boldsymbol{y} - \boldsymbol{A}\boldsymbol{x}\|_{\boldsymbol{\Sigma}}^2$ (up to an additive constant that does not depend on $\boldsymbol{x}$ and $\boldsymbol{y}$) where $\|\cdot\|_{\boldsymbol{\Sigma}}^2 := \langle\cdot, \boldsymbol{\Sigma}^{-1}\cdot\rangle$. It is then straightforward to show that

$$\pi^{Z|X=\boldsymbol{x}} = \mathcal{N}(\boldsymbol{m}(\boldsymbol{x}), \boldsymbol{\Lambda}^{-1})$$

where $\boldsymbol{\Lambda} := \boldsymbol{A}^T\boldsymbol{\Sigma}^{-1}\boldsymbol{A} + \frac{1}{\rho^2}\boldsymbol{I}$ and $\boldsymbol{m}(\boldsymbol{x}) := \boldsymbol{\Lambda}^{-1}(\boldsymbol{A}^T\boldsymbol{\Sigma}^{-1}\boldsymbol{y} + \frac{1}{\rho^2}\boldsymbol{x})$. The problem of sampling from Gaussian distributions has been systematically studied [72]. We refer readers to Appendix C.1 for a more detailed discussion.

**General case**  For general nonlinear inverse problems, the likelihood step is not sampling from a Gaussian distribution anymore. Nevertheless, since we have access to $\pi^{Z|X=\boldsymbol{x}}$ in closed form up to a multiplicative factor, we can use Monte Carlo methods based on Langevin dynamics to draw samples from it as long as the likelihood potential is differentiable. Specifically, we first set up the following Langevin SDE that admits $\pi^{Z|X=\boldsymbol{x}}$ as the stationary distribution

$$\mathrm{d}\boldsymbol{z}_t = \nabla \log \pi^{Z|X=\boldsymbol{x}}(\boldsymbol{z}_t)\mathrm{d}t + \sqrt{2}\mathrm{d}\boldsymbol{w}_t = \left[-\nabla f(\boldsymbol{z};\boldsymbol{y}) - \frac{1}{\rho^2}(\boldsymbol{z} - \boldsymbol{x})\right]\mathrm{d}t + \sqrt{2}\mathrm{d}\boldsymbol{w}_t.$$

We then initialize the SDE at $\boldsymbol{z}_0 = \boldsymbol{x}$ and run it with Euler discretization. The pseudocode is provided in Appendix C.1.

### 3.2 Prior step: denoising via the EDM framework

For the prior step at iteration $k$, we sample

$$\boldsymbol{x}^{(k+1)} \sim \pi^{X|Z=\boldsymbol{z}^{(k)}}(\boldsymbol{x}) \propto \exp\left(-g(\boldsymbol{x}) - \frac{1}{2\rho^2}\|\boldsymbol{x} - \boldsymbol{z}^{(k)}\|_2^2\right). \tag{6}$$

A closer examination of (6) reveals that this prior step is essentially to draw posterior samples for a Gaussian denoising problem, where the "measurement" is $\boldsymbol{z}^{(k)}$, the noise level is $\rho$, and the prior distribution is $p(\boldsymbol{x}) \propto \exp(-g(\boldsymbol{x}))$.

We tackle this denoising posterior sampling problem within SGS using DMs as image priors. In particular, we leverage the EDM framework [37], which was originally proposed to unify various

formulations of DMs for unconditional image generation. To see the connection of the EDM framework to (6), consider a family of mollified distributions $p(\boldsymbol{x}; \sigma)$ given by adding i.i.d Gaussian noise of standard deviation $\sigma$ to the prior distribution $p(\boldsymbol{x})$, i.e. $\boldsymbol{x} + \sigma\boldsymbol{\epsilon} \sim p(\boldsymbol{x}; \sigma)$. The core idea of the EDM framework is that a variety of state-of-the-art DMs can be unified into the following reverse SDE:

$$\mathrm{d}\boldsymbol{x}_t = \left[ \frac{\dot{s}(t)}{s(t)}\boldsymbol{x}_t - 2s(t)^2\dot{\sigma}(t)\sigma(t)\nabla\log p\left(\frac{\boldsymbol{x}_t}{s(t)}; \sigma(t)\right) \right]\mathrm{d}t + s(t)\sqrt{2\dot{\sigma}(t)\sigma(t)}\mathrm{d}\bar{\boldsymbol{w}}_t \qquad (7)$$

where $\bar{\boldsymbol{w}}_t$ is an $n$-dimensional Wiener process running backward in time, $\sigma(t) > 0$ is a pre-defined noise level schedule with $\sigma(0) = 0$, $s(t)$ is a pre-defined scaling schedule, and $\dot{\sigma}(t)$, $\dot{s}(t)$ are their time derivatives. As shown in [37], the defining property of (7) is that $\boldsymbol{x}_t/s(t) \sim p(\boldsymbol{x}; \sigma(t))$ for any time $t$. Therefore, solving this SDE backward in time allows us to travel from any noise level $\sigma(t)$ to the clean image distribution at $t = 0$. This means that we can use (7) to solve (6) with arbitrary noise level $\rho$ as long as $\rho$ is within the range of $\sigma(t)$. Indeed, the distribution of $\boldsymbol{x}_0$ conditioned on $\boldsymbol{x}_t$ is

$$p(\boldsymbol{x}_0|\boldsymbol{x}_t) \propto p(\boldsymbol{x}_t|\boldsymbol{x}_0)p(\boldsymbol{x}_0) \propto \mathcal{N}(s(t)\boldsymbol{x}_0, s(t)^2\sigma(t)^2\boldsymbol{I}) \exp(-g(\boldsymbol{x}_0))$$

$$\propto \exp\left(-g(\boldsymbol{x}_0) - \frac{1}{2\sigma(t)^2}\|\boldsymbol{x}_0 - \boldsymbol{x}_t/s(t)\|_2^2\right).$$

We highlight that the last line exactly matches (6) when $\boldsymbol{x}_t = s(t)\boldsymbol{z}^{(k)}$ and $\sigma(t) = \rho$. Therefore, we can naturally design a practical algorithm that samples (6) by following these three steps: (1) find $t^*$ such that $\sigma(t^*) = \rho$, (2) initialize at $\boldsymbol{x}_{t^*} = s(t^*)\boldsymbol{z}^{(k)}$, and (3) solve (7) backward from $t^*$ to 0 by choosing the discretization time steps and integration scheme. Through this unified interface, any DMs, once converted to the EDM formulation, can be directly turned into a rigorous solver for (6).

Leveraging the connection with EDM, our prior step implementation comes with a large design space that encompasses a variety of existing DMs, such as DDPM (or VP-SDE) [33], VE-SDE [63], and iDDPM [52]. In our experiments, we conduct posterior sampling with all these different models within our framework and all of them provide high-quality samples. The pseudocode of our implementation and more details on the EDM formulation for the prior step is given in Appendix C.2.

## 3.3 Putting it all together

The pseudocode of PnP-DM in complete form is presented in Algorithm 1. PnP-DM alternates between the two sampling steps with an annealing schedule $\{\rho_k\}$ for the coupling parameter. We find that the annealing schedule on $\rho$ accelerates the mixing time of the Markov chain and prevents the algorithm from getting stuck in bad local minima for solving highly ill-posed inverse problems. This is a common practice in both Langevin-based [40, 34, 65] and SGS-based [6, 78] MCMC algorithms to improve the empirical performance in solving inverse problems.

Our work shares some similarities with PnP-SGS [19] but contains three main key differences. First, as demonstrated in our experiments, we investigate three nonlinear inverse problems, while nonlinear inverse problems are beyond the scope of [19]. Our experiments show that PnP-SGS struggles with challenging nonlinear inverse problems such as Fourier phase retrieval. Second, we adopt the EDM formulation to ensure that the prior step of PnP-DM is a rigorous mapping from the image manifold with the desired noise level to the clean image manifold, aligning with the theory of SGS. In contrast, the prior step of PnP-SGS [19] is heuristic (which is also pointed out by [78]) and not rigorously designed to sample (6). Third, unlike PnP-SGS [19] that uses a constant $\rho$, we consider an annealing schedule $\{\rho_k\}$ for the coupling parameter, which is important for highly ill-posed inverse problems.

---

**Algorithm 1** Plug-and-Play Diffusion Models (PnP-DM)

---

**Input:** initialization $\boldsymbol{x}_0 \in \mathbb{R}^n$, total number of iterations $K > 0$, coupling strength schedule $\{\rho_k > 0\}_{k=0}^{K-1}$, likelihood potential $f(\,\cdot\,; \boldsymbol{y})$ with measurements $\boldsymbol{y} \in \mathbb{R}^m$, pretrained model $D_\theta(\,\cdot\,; \,\cdot\,)$ that approximates $\nabla\log p(\boldsymbol{x}; \sigma)$ with $(D_\theta(\boldsymbol{x}; \sigma) - \boldsymbol{x})/\sigma^2$.
1: **for** $k = 0, ..., K - 1$ **do**
2:     $\boldsymbol{z}^{(k)} \leftarrow \texttt{LikelihoodStep}(\boldsymbol{x}^{(k)}, \rho_k, f(\,\cdot\,; \boldsymbol{y}))$                $\triangleright$ Section 3.1
3:     $\boldsymbol{x}^{(k+1)} \leftarrow \texttt{PriorStep}(\boldsymbol{z}^{(k)}, \rho_k, D_\theta(\,\cdot\,; \,\cdot\,))$           $\triangleright$ Section 3.2
4: **end for**
5: **return** $\boldsymbol{x}^{(k+1)}$

---

## 3.4 Theoretical insights

We provide some theoretical insights on the non-asymptotic behavior of PnP-DM. We start with the following definitions. For two probability measures $\mu$ and $\widetilde{\mu}$ such that $\mu \ll \widetilde{\mu}$, the *Kullback–Leibler (KL) divergence* and *Fisher information (or Fisher divergence)* of $\mu$ with respect to $\widetilde{\mu}$ are defined, respectively, as

$$\mathsf{KL}(\mu||\widetilde{\mu}) := \int \mu \log \frac{\mu}{\widetilde{\mu}} \quad \text{and} \quad \mathsf{FI}(\mu||\widetilde{\mu}) := \int \mu \left\| \nabla \log \frac{\mu}{\widetilde{\mu}} \right\|_2^2 .$$

Both divergences are equal to zero if and only if $\mu = \widetilde{\mu}$. KL divergence is a common metric for quantifying the difference of one distribution with respect to another. Fisher information has been used for analyzing the stationarity of sampling algorithms [3, 66].

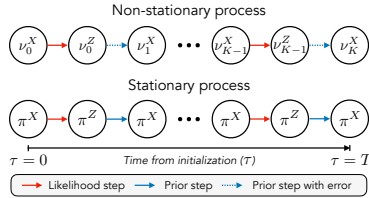

Figure 3: A conceptual illustration of the non-stationary and stationary time-continuous processes as interpolations of $K$ discretize iterations of PnP-DM.

We analyze PnP-DM via a continuous-time perspective, leveraging the interpolation techniques introduced for Langevin Monte Carlo [68, 3, 66]. We assume that the likelihood step (5) can be implemented exactly and the prior step (6) involves running the reverse diffusion process (7) with an approximated score function $s_t \approx \nabla \log p_t := \nabla \log p(\cdot; \sigma(t))$. Let $\nu_0^X$ be the distribution of the initialization $x^{(0)}$. Let $\nu_k^Z$ and $\nu_{k+1}^X$ be the distributions of $z^{(k)}$ and $x^{(k+1)}$ at the $k^{\text{th}}$ iteration. Recall that the stationary distributions are $\pi^X$ and $\pi^Z$. Our analysis is concerned with two *continuous-time* processes: (1) the non-stationary process from $\nu_0^X$, a non-stationary initialization, to $\nu_K^X$ where (7) is run with the approximated score function $s_t$ and (2) the stationary process that alternates between stationary distributions $\pi^X$ and $\pi^Z$. These two processes are the interpolation PnP-DM in non-stationary and stationary states and define continuous transitions over discrete iterations. A conceptual illustration of the two processes is provided in Figure 3 with the exact formulations in Appendix A. Now we present our main result:

**Theorem 3.1.** *Consider running $K$ iterations of PnP-DM with $\rho_k \equiv \rho > 0$ and a score estimate $s_t \approx \nabla \log p_t := \nabla \log p(\cdot; \sigma(t))$. Let $t^* > 0$ be such that $\sigma(t^*) = \rho$ and $\delta := \inf_{t \in [0, t^*]} v(t)$ where $v(t) := s(t)\sqrt{2\dot{\sigma}(t)\sigma(t)}$. Define $\nu_\tau$ and $\pi_\tau$ as the distributions at time $\tau$ of the non-stationary and stationary process, respectively. Then, for over $K$ iterations of PnP-DM, or equivalently over $\tau \in [0, T]$ with $T := K(t^* + 1)$, we have*

$$\frac{1}{T} \int_0^T \mathsf{FI}(\pi_\tau||\nu_\tau) \, \mathrm{d}\tau \le \underbrace{\frac{4\mathsf{KL}(\pi^X||\nu_0^X)}{K(t^* + 1)\min(\rho, \delta)^2}}_{\textit{convergence from initialization}} + \underbrace{\frac{4\epsilon_{score}}{(t^* + 1)\delta^2}}_{\textit{score approximation error}} , \qquad (8)$$

*where we assume that the score estimation error $\epsilon_{score} := \int_1^{t^*+1} v(\tau)^2 \mathbb{E}_{\pi_\tau} \|s_\tau - \nabla \log p_\tau\|_2^2 \mathrm{d}\tau < \infty$.*

The proof is provided in Appendix A. This theorem states that the average distance (measured by Fisher information) of the non-stationary process with respect to the stationary process over $K$ iterations of PnP-DM goes to zero at a rate of $O(1/K)$ under certain conditions up to the score approximation error. Note that our theory only requires $L^2$-accurate score estimate under the measure $\pi_\tau$, which is a relatively weaker condition than the common $L^\infty$-accurate score estimate assumption in prior analysis of sampling methods involving score estimates [5, 66]. This result resembles the first-order stationarity for Langevin Monte Carlo [3]. Unlike the non-asymptotic analysis in [78], we utilize the average Fisher information instead of the total variation distance, enabling us to obtain an explicit convergence rate. Here $\delta$ is the infimum of the diffusion coefficient along the reverse diffusion in (7); see further discussions on the role of $\delta$ in Appendix A.3. Our theory shows that the accurate implementations of the two sampling steps lead to a sampler that provably converges to the stationary process that alternates between the two target stationary distributions.

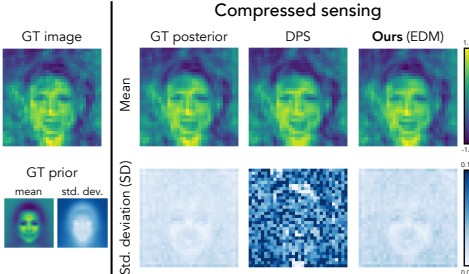

Figure 4: Results on a synthetic problem with the ground truth posterior available. PnP-DM can sample it more accurately that DPS [17].

Table 1: Quantitative comparison on three noisy linear inverse problems for 100 FFHQ color test images. **Bold:** best; Underline: second best.

| Method | Gaussian deblur | | | Motion deblur | | | Super-resolution (4×) | | |
|---|---|---|---|---|---|---|---|---|---|
| | PSNR (↑) | SSIM (↑) | LPIPS (↓) | PSNR (↑) | SSIM (↑) | LPIPS (↓) | PSNR (↑) | SSIM (↑) | LPIPS (↓) |
| PnP-ADMM [13] | 26.88 | 0.7855 | 0.3472 | 26.55 | 0.7655 | 0.3600 | 26.61 | 0.7634 | 0.3766 |
| DPIR [80] | 28.74 | 0.8348 | 0.2677 | 29.97 | 0.8529 | 0.2404 | 28.75 | 0.8378 | 0.2577 |
| DDRM [39] | 27.05 | 0.7819 | 0.2570 | – | – | – | 29.47 | **0.8437** | 0.2322 |
| DPS [17] | 28.83 | 0.8212 | 0.2330 | 27.87 | 0.8035 | 0.2542 | 29.45 | 0.8379 | 0.2274 |
| PnP-SGS [19] | 27.46 | 0.8356 | 0.2445 | 28.98 | 0.8447 | 0.2190 | 28.30 | 0.8349 | 0.2160 |
| DPnP [78] | 29.24 | 0.8360 | 0.2098 | 30.21 | 0.8527 | 0.2010 | 29.32 | 0.8407 | 0.2127 |
| PnP-DM (VP) | 29.46 | 0.8215 | 0.2202 | 30.06 | 0.8336 | 0.2099 | 29.40 | 0.8238 | 0.2219 |
| PnP-DM (VE) | 29.65 | 0.8399 | **0.2090** | **30.38** | 0.8547 | **0.1971** | 29.57 | 0.8431 | **0.2108** |
| PnP-DM (iDDPM) | 29.60 | 0.8383 | 0.2203 | 30.26 | 0.8507 | 0.2103 | 29.53 | 0.8404 | 0.2213 |
| PnP-DM (EDM) | **29.66** | **0.8411** | 0.2170 | 30.35 | **0.8547** | 0.2062 | **29.60** | 0.8435 | 0.2191 |

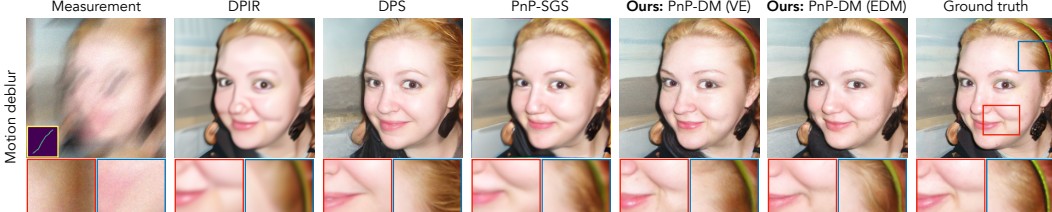

Figure 5: Visual examples for the motion deblur problem ($\sigma_{\boldsymbol{y}} = 0.05$). We visualize one sample generated by each sampling algorithm.

## 4 Experiments

### 4.1 Validation with ground truth posterior

We first demonstrate the accuracy of PnP-DM for posterior sampling on a simulated compressed sensing problem with a Gaussian prior where the posterior distribution can be expressed in a closed form. The mean and per-pixel standard deviation of the prior are visualized on the bottom left of Figure 4. The linear forward model $\boldsymbol{A} \in \mathbb{R}^{m \times n}$ is a Gaussian matrix ($m = n/2$), i.e. $\boldsymbol{A}_{ij} \sim \mathcal{N}(0, 1)$. A test image is randomly generated from the prior (see top left of Figure 4), and the measurement is calculated according to (1) with $\boldsymbol{n} \sim \mathcal{N}(\boldsymbol{0}, 0.01^2 \boldsymbol{I})$. We compare our method with the popular DM-based method DPS [17]. We draw 1,000 samples and visualize the empirical mean and per-pixel standard deviation for both algorithms. Compared with the true posterior (second column), we find that the both methods accurately estimate the mean. However, the standard deviation image estimated by DPS significantly deviates from the ground truth. In contrast, our standard deviation image matches the ground truth in terms of both absolute magnitude and spatial distribution. These results highlight the accuracy of our method over DPS by taking a more principled Bayesian approach.

### 4.2 Benchmark experiments

**Dataset and inverse problems** We test our proposed algorithm and several baseline methods on 100 images from the validation set of the FFHQ dataset [38] for five inverse problems: (1) *Gaussian deblur* with kernel size 61×61 and standard deviation 3.0, (2) *Motion deblur* with kernel size 61×61 and intensity of 0.5, (3) *Super-resolution* with 4× downsampling ratio, (4) the coded diffraction patterns (CDP) reconstruction problem (nonlinear) in [10, 51] (phase retrieval with a phase mask), and (5) the Fourier phase retrieval (nonlinear) with 4× oversampling. We add i.i.d. Gaussian noise to all the simulated measurements $\boldsymbol{y}$. In particular, i.e. $\boldsymbol{n} \sim \mathcal{N}(\boldsymbol{0}, \sigma_{\boldsymbol{y}}^2 \boldsymbol{I})$. For all problems except for Fourier phase retrieval, the noise standard deviation is set as $\sigma_{\boldsymbol{y}} = 0.05$. Due to the severe ill-posedness of Fourier phase retrieval, we consider a smaller noise standard deviation $\sigma_{\boldsymbol{y}} = 0.01$.

**Baselines and comparison protocols** We consider four variants of DMs as plug-in priors for our method, namely VP-SDE (VP) [33], VE-SDE (VE) [63], iDDPM [52], and EDM [37]. We compare our method with various baselines, including (1) optimization-based methods: PnP-ADMM [13], DPIR [80]; (2) conditional DMs: DDRM [39], DPS [18]; and (3) SGS-based method: PnP-SGS [19], DPnP [78]. For fair comparison, we use the same pre-trained score function checkpoint for all DM-based methods. Since the pre-trained score function was trained with the DDPM formu-

lation (VP-SDE) [33], we convert it to the EDM formulation by applying the VP preconditioning [37]. We use the Peak Signal-to-Noise Ratio (PSNR), the Structural Similarity Index Measure (SSIM), and the Learned Perceptual Image Patch Similarity (LPIPS) distance for quantitative comparison. For each sampling method, we draw 20 randoms samples, calculate their mean, and report the metrics on the mean image. More experimental details are provided in Appendices B, C, D.

**Results: linear problems** A quantitative comparison is provided in Table 1. PnP-DM generally outperforms the baseline methods and that the VE and EDM variants consistently outperform the other two variants on these linear problems. Figure 5 contains visual examples for the motion deblur problem (see Appendix F.2 for the other two linear problems). PnP-DM provides high-quality reconstructions that are both sharp and consistent with the ground truth image. We also provide an uncertainty quantification analysis based on pixel-wise statistics in Figure 6. In the left three columns, we visualize the absolute error ($|\bar{\boldsymbol{x}} - \boldsymbol{x}|$), standard deviation (std), and absolute z-score ($|\bar{\boldsymbol{x}} - \boldsymbol{x}|$/std). In the third column, red pixels highlight locations where the ground truth pixel values are outliers of the 3-sigma credible interval (CI) under the estimated posterior uncertainty. The fourth column contains scatter plots of $|\bar{\boldsymbol{x}} - \boldsymbol{x}|$ versus std for each pixel of the reconstructions, where red boxes show the percentages of outliers (outside of 3-sigma CI) and gray boxes indicate the percentages within the 3-sigma CI. Similar to the

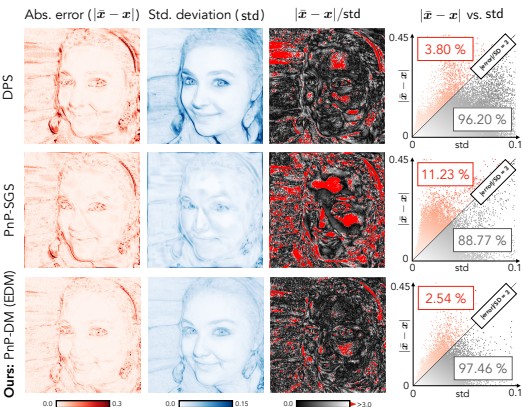

Figure 6: Comparison of uncertainty quantification (UQ) for the motion deblur. Left 3 columns: absolute error ($|\bar{\boldsymbol{x}} - \boldsymbol{x}|$), standard deviation (std), and absolute z-score ($|\bar{\boldsymbol{x}} - \boldsymbol{x}|$/std) with the outlier pixels in red. Right column: scatter plot of $|\bar{\boldsymbol{x}} - \boldsymbol{x}|$ versus std. Note that PnP-DM leads to a better UQ performance than the baselines by having the lowest percentage of outliers while avoiding having overestimated per-pixel standard deviations.

synthetic prior experiment, DPS tends to have larger standard deviation estimations, as shown by the less concentrated distribution of gray points around the origin. Compared with baselines, especially PnP-SGS, our approach captures a higher percentage (97.46%) of ground truth pixels than the baselines (96.20% and 88.77%). If the true posterior were truly Gaussian, 99% of the ground-truth pixels should lie within the 3-sigma CI; however, as the posterior is not Gaussian with a DM-based prior, we do not necessarily expect to reach 99% coverage.

**Results: nonlinear problems** We provide a quantitative comparison in Table 2. For the CDP reconstruction problem, PnP-DM performs on par with DPS but outperforms other SGS-based methods. We then consider the Fourier phase retrieval (FPR) problem, which is known to be a challenging nonlinear inverse problem. One challenge lies in its invariance to 180° rotation, so the posterior distribution have two modes, one with upright images and another with 180°-rotated images, that equally fit the measurement. To increase the chance of getting properly-oriented reconstructions, we run each algorithm with four different random initializations and report the metrics for the best run, following the practice in [18]. We find that PnP-DM significantly outperforms the baselines on this highly ill-posed inverse problem. As shown in Figure 7 (a), our method can provide high-quality reconstructions for both orientations, while the baseline methods fail to capture at least one of the two modes. We further run our method for a test image with 100 different random initialization and collect reconstructions in both orientations that are above 28dB in PSNR (90 out of 100 runs). The percentage of upright and rotated reconstructions are visualized by the pie chart in Figure 7 (b). With a prior on upright face images, our method generate mostly samples with the upright orientation. Nevertheless, it can also find the other mode that has an equal likelihood, demonstrating its ability to capture multi-modal posterior distributions.

### 4.3 Experiments on black hole imaging

**Problem setup** We finally validate PnP-DM on a real-world nonlinear imaging inverse problem: black hole imaging (BHI) (see Appendix B for more details). A visual illustration of BHI is provided

Table 2: Quantitative evaluation on two noisy nonlinear inverse problems for 100 FFHQ grayscale test images. **Bold:** best; Underline: second best.

| Method | Coded diffraction patterns | | | Fourier phase retrieval | | |
|---|---|---|---|---|---|---|
| | PSNR (↑) | SSIM (↑) | LPIPS (↓) | PSNR (↑) | SSIM (↑) | LPIPS (↓) |
| HIO [29] | – | – | – | 20.66 | 0.4308 | 0.6469 |
| DPS [17] | **33.43** | 0.9049 | **0.1374** | 23.60 | 0.6804 | 0.3126 |
| PnP-SGS [19] | 32.19 | 0.8889 | 0.2010 | 15.36 | 0.3659 | 0.5730 |
| DPnP [78] | 32.19 | 0.8853 | 0.2000 | 29.28 | 0.8397 | 0.2180 |
| PnP-DM (VP) | 32.91 | 0.8846 | 0.1906 | 30.36 | 0.8553 | 0.2115 |
| PnP-DM (VE) | 33.13 | 0.8971 | 0.1663 | 29.88 | 0.8464 | 0.2186 |
| PnP-DM (iDDPM) | 33.35 | **0.9083** | 0.1471 | 30.61 | 0.8718 | **0.1975** |
| PnP-DM (EDM) | 33.25 | 0.9050 | 0.1386 | **31.14** | **0.8731** | 0.2024 |

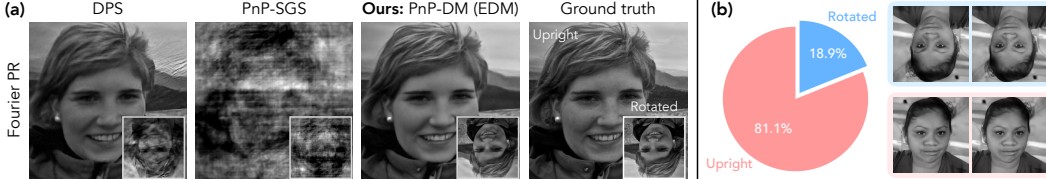

Figure 7: Results of the Fourier phase retrieval problem. (a) PnP-DM provides both upright and rotated reconstructions (two modes given by the invariance of the forward model to 180° rotation) with high fidelity, while the baseline methods cannot. (b) We visualize the percentages of upright and rotated reconstructions out of 90 runs for a test image with two samples for each orientation.

in Figure 8 (a). This BHI inverse problem is severely ill-posed. Even with an Earth-sized telescope, only a small fraction of the Fourier frequencies of the target black hole can be measured (region within the red box); in reality, this region is further subsampled with a highly sparse pattern (black lines). Additionally, the atmospheric noise causes nonlinearity of this BHI problem that sometimes results in a multi-modal posterior distribution of the reconstructed image [64]. Here we demonstrate the effectiveness of PnP-DM in capturing a multi-modal posterior distribution. For brevity, we restrict our choice of diffusion models in PnP-DM to EDM and use DPS as the baseline.

**Results on simulated data** We use the simulated data from [64] where the measurements are generated assuming that the ground-truth black hole image were at the location of the Sagittarius A* black hole. Figure 8 (b) visually compares the results obtained by PnP-DM and DPS. We use the t-SNE method [67] to cluster the generated samples (100 for each method) and identify two modes in the samples generated by PnP-DM and three modes in those generated by DPS. We visualize the mean and three samples for each image mode. A metric for quantifying the degree of data mismatch is labeled on the top right corner of each image. As illustrated by both the mean and sample images, PnP-DM successfully captures the two modes previously identified for this dataset [64]. Note that PnP-DM generates high-fidelity samples from both modes with sharp details of the flux ring, and its samples from "Mode 1" align well with the ground truth image. In contrast, two out of the three modes sampled by DPS fail to exhibit a meaningful black hole structure and do not correspond with the observed measurements, as indicated by the significantly larger data mismatch values.

**Results on real data** Finally, we apply PnP-DM to the real M87 black hole data from April 6th, 2017 [21], with the results shown in Figure 1. By leveraging an expressive DM-based image prior, PnP-DM generates high-quality samples that are both visually plausible and consistent with the ring diameters observed in the official EHT reconstruction. These results highlight the robustness and effectiveness of our method in tackling a highly ill-posed real-world inverse problem.

## 5 Conclusion

We have introduced PnP-DM, a posterior sampling method for solving imaging inverse problems. The backbone of our method is a split Gibbs sampler that iteratively alternates between two steps that separately involve the likelihood and prior. Crucially, we establish a link between the prior step and a general DM framework known as the EDM formulation. By leveraging this connection, we seamlessly integrate a diverse range of state-of-the-art DMs as priors through a unified interface. Experimental results demonstrate that our method outperforms existing DM-based methods across

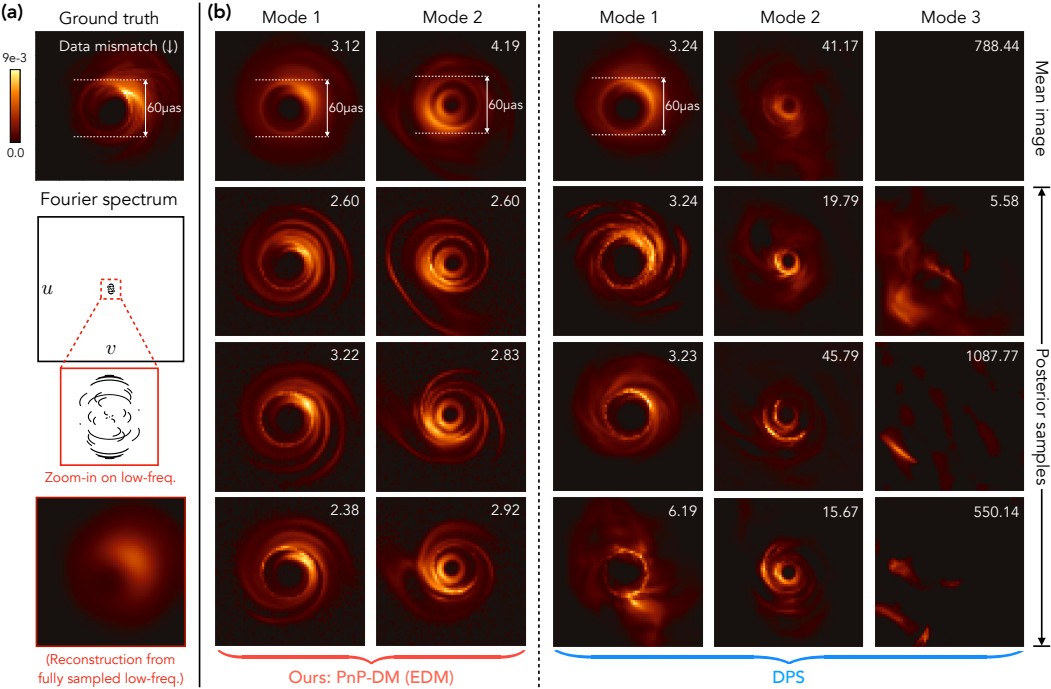

Figure 8: Results on a nonlinear and severely ill-posed black hole imaging problem. Our method, PnP-DM, is compared with the conditional diffusion model baseline DPS. A metric quantifying the mismatch with the observed measurements is labeled for each sample, which should be around 2 for ideal measurement fit. Samples generated by PnP-DM exhibit two distinct modes with sharp details and a consistent ring structure, while samples given by DPS display inconsistent ring sizes and sometimes fail to capture the black hole structure entirely with samples having poor measurement fit.

both linear and nonlinear inverse problems, including a nonlinear and severely ill-posed black hole interferometric imaging problem.

**Limitations**    PnP-DM can be further improved in the following two aspects. First, PnP-DM currently requires evaluating the likelihood and prior steps for the entire image at a time. This potentially poses computational challenges in solving large-scale inverse problems (e.g. 3D imaging) or those with expensive likelihood evaluation (e.g. PDE inverse problems). Second, the current theoretical analysis does not consider the approximation error introduced in the likelihood step for general nonlinear inverse problems when running Langevin MCMC for finite iterations. Explicit incorporation of this error would offer further insights into the empirical performance of PnP-DM.

**Broader impacts**    We expect this work to make a positive impact in computational imaging and related application domains. For many imaging problems, there is a need to facilitate image reconstruction with expressive image priors and quantify uncertainty, which could lead to better imaging systems that enables further understanding of the imaging target. Nonetheless, as we are introducing DMs as priors into the imaging process, it is inevitable to inherent the potential bias of these models.

# 6    Acknowledgments

The authors thank Charles Gammie, Ben Prather, Abhishek Joshi, Vedant Dhruv, and Chi-kwan Chan for providing the black hole simulations. The authors also thank the generous funding from Schmidt Sciences and the Heritage Medical Research Fellowship. Z.W. was supported by an Amazon AI4Science Fellowship. Y.S. was supported by a Computing, Data, and Society Fellowship. B.Z. was supported by a Kortschak Fellowship.

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

# A Theory

## A.1 Interpolation of PnP-DM

In this section, we formally introduce the interpolation of PnP-DM. We consider the case where the coupling strength $\rho$ is constant, i.e. $\rho_k \equiv \rho$ and make the following assumption.

**Assumption A.1.** *There exists a unique $t^*$ such that $\sigma(t^*) = \rho$.*

This assumption is satisfied for common diffusion models. Popular choices of the noise level schedule include $\sigma(t) = t$ or $\sigma(t) = \sqrt{t}$, which are monotonically increasing functions of $t$. We first present two propositions showing that the two steps in SGS can be implemented by running two SDEs.

**Proposition A.2** (Brownian bridge for the likelihood step). *For iteration $k$ with iterate $\boldsymbol{x}^{(k)}$, the likelihood step of SGS is equivalent to solving the following SDE from $t = 0$ to $t = 1$:*

$$\mathrm{d}\boldsymbol{x}_t = \rho^2 \nabla \log \phi_t(\boldsymbol{x}_t)\mathrm{d}t + \rho \mathrm{d}\boldsymbol{w}_t \tag{9}$$

*where $\boldsymbol{x}_0 = \boldsymbol{x}^{(k)}$ and $\phi_t(\boldsymbol{x}) := \int \exp[-f(\boldsymbol{z};\boldsymbol{y}) - \frac{1}{2\rho^2(1-t)}\|\boldsymbol{x} - \boldsymbol{z}\|_2^2]\mathrm{d}\boldsymbol{z}$.*

*Proof.* This proposition is due to the Brownian bridge construction presented in Lemma 4 of [79]. This SDE satisfies that $p(\boldsymbol{x}_1|\boldsymbol{x}_0) \propto \exp\left(-f(\boldsymbol{x}_1;\boldsymbol{y}) - \frac{1}{2\rho^2}\|\boldsymbol{x}_0 - \boldsymbol{x}_1\|_2^2\right)$. Therefore, solving (9) from $t = 0$ to $t = 1$ is equivalent to taking a likelihood step. □

**Proposition A.3** (EDM reverse diffusion for the prior step). *For iteration $k$ with iterate $\boldsymbol{z}^{(k)}$, the prior step of SGS is equivalent to solving the following SDE from $t = t^*$ to $t = 0$:*

$$\mathrm{d}\boldsymbol{x}_t = \left[u(t)\boldsymbol{x}_t - v(t)^2 \nabla \log p_t(\boldsymbol{x}_t)\right]\mathrm{d}t + v(t)\mathrm{d}\bar{\boldsymbol{w}}_t \tag{10}$$

*where $\boldsymbol{x}_{t^*} = s(t^*)\boldsymbol{z}^{(k)}$, $u(t) := \frac{\dot{s}(t)}{s(t)}$, $v(t) := s(t)\sqrt{2\dot{\sigma}(t)\sigma(t)}$, and $p_t$ is the distribution of $s(t)\boldsymbol{x} + s(t)\sigma(t)\boldsymbol{\epsilon}$ with $\boldsymbol{x}$ following the prior distribution $p(\boldsymbol{x}) \propto \exp(-g(\boldsymbol{x}))$ and $\boldsymbol{\epsilon} \sim \mathcal{N}(\boldsymbol{0},\boldsymbol{I})$.*

*Proof.* First note that (10) is exactly (7) written in terms of $u(t)$ and $v(t)$. We know that the (10) is the reverse SDE of the following SDE

$$\mathrm{d}\boldsymbol{x}_t = u(t)\boldsymbol{x}_t\mathrm{d}t + v(t)\mathrm{d}\boldsymbol{w}_t. \tag{11}$$

where $\boldsymbol{x}_0 \sim p(\boldsymbol{x})$ and $p_t$ is the marginal distribution of $\boldsymbol{x}_t$. As we showed in the main text, it holds for (11) that

$$p(\boldsymbol{x}_0|\boldsymbol{x}_t) \propto \exp\left(-g(\boldsymbol{x}_0) - \frac{1}{2\sigma(t)^2}\|\boldsymbol{x}_0 - \boldsymbol{x}_t/s(t)\|_2^2\right).$$

As (10) is the time-reversed process of (11), they share the same path distribution and thus the same conditional distribution $p(\boldsymbol{x}_0|\boldsymbol{x}_t)$. So, if we set $\boldsymbol{x}_{t^*} = s(t^*)\boldsymbol{z}^{(k)}$, we have that

$$p(\boldsymbol{x}_0|\boldsymbol{x}_{t^*}) \propto \exp\left(-g(\boldsymbol{x}_0) - \frac{1}{2\sigma(t^*)^2}\|\boldsymbol{x}_0 - \boldsymbol{z}^{(k)}\|_2^2\right) \propto \exp\left(-g(\boldsymbol{x}_0) - \frac{1}{2\rho^2}\|\boldsymbol{x}_0 - \boldsymbol{z}^{(k)}\|_2^2\right),$$

which is the desired conditional distribution of the prior step. Therefore, solving (10) from $t = t^*$ to $t = 0$ is equivalent to taking a prior step. □

Due to Proposition A.2 and Proposition A.3, the SDEs (9) and (10) implement the two desired conditional distributions in SGS. In PnP-DM, the prior step involves a network that approximates the score function of the prior distribution, i.e. $\boldsymbol{s}_t \approx \nabla \log p_t$, so the continuous-time process for the actual update is

$$\mathrm{d}\boldsymbol{x}_t = \left[u(t)\boldsymbol{x}_t - v(t)^2 \boldsymbol{s}_t(\boldsymbol{x}_t)\right]\mathrm{d}t + v(t)\mathrm{d}\bar{\boldsymbol{w}}_t. \tag{12}$$

We can then interpolate PnP-DM by considering a dynamic that alternates between running (9) and (12).

Since each likelihood step takes 1 unit of time and each prior step takes $t^*$ unit of time, the total time of the interpolating process for $K$ iterations of PnP-DM is $T := K(t^* + 1)$. We use $\tau$ to denote the time that has elapsed from initializing PnP-DM with $\boldsymbol{x}^{(0)}$. We define $\{\nu_\tau\}$ and $\{\pi_\tau\}$ as the distributions at time $\tau$ of the non-stationary process initialized at $\boldsymbol{x}^{(0)} \sim \nu_0^X$ (Figure 3 top) and the stationary process initialized at $\boldsymbol{x}^{(0)} \sim \pi^X$ (Figure 3 bottom), respectively. Therefore, we have

- $\nu_\tau = \nu_k^X$, $\pi_\tau = \pi^X$ for $\tau = k(t^* + 1)$ with $k = 0, \cdots, K$, and
- $\nu_\tau = \nu_k^Z$, $\pi_\tau = \pi^Z$ for $\tau = k(t^* + 1) + 1$ with $k = 0, \cdots, K - 1$.

## A.2 Proof of Theorem 3.1

Before proving our main result, we present a key lemma for our analysis, which quantifies the time-derivative of the KL divergence in terms of the Fisher information along a pair of general diffusion processes.

**Lemma A.4.** *Given the following pair of diffusion processes*

$$\mathrm{d}\boldsymbol{x}_t = b(\boldsymbol{x}_t, t)\mathrm{d}t + c(t)\mathrm{d}\boldsymbol{w}_t \tag{13}$$

$$\mathrm{d}\widetilde{\boldsymbol{x}}_t = \widetilde{b}(\widetilde{\boldsymbol{x}}_t, t)\mathrm{d}t + c(t)\mathrm{d}\boldsymbol{w}_t \tag{14}$$

*where $b(\cdot, \cdot) : \mathbb{R}^n \times \mathbb{R} \to \mathbb{R}^n$, $\widetilde{b}(\cdot, \cdot) : \mathbb{R}^n \times \mathbb{R} \to \mathbb{R}^n$, and $c(\cdot) : \mathbb{R} \to \mathbb{R}$. Let $\mu_t$ be the distribution of $\boldsymbol{x}_t$ initialized with $\boldsymbol{x}_0 \sim \mu_0$ for (13), and let $\widetilde{\mu}_t$ be the distribution of $\widetilde{\boldsymbol{x}}_t$ initialized with $\widetilde{\boldsymbol{x}}_0 \sim \widetilde{\mu}_0$ for (14). Then we have*

$$\partial_t \mathsf{KL}(\mu_t || \widetilde{\mu}_t) \leq -\frac{c(t)^2}{4} \mathsf{FI}(\mu_t || \widetilde{\mu}_t) + \frac{1}{c(t)^2} \int \left\| \widetilde{b}_t - b_t \right\|_2^2 \mu_t. \tag{15}$$

*Proof of Lemma A.4.* Writing $b(\cdot, t)$ as $b_t$ and $\widetilde{b}(\cdot, t)$ as $\widetilde{b}_t$, by the Fokker-Planck equations of (13) and (14), we have that

$$\partial_t \mu_t = \mathrm{div}\left[ \left( \frac{c(t)^2}{2} \nabla \log \mu_t - b_t \right) \mu_t \right] \quad \text{and} \quad \partial_t \widetilde{\mu}_t = \mathrm{div}\left[ \left( \frac{c(t)^2}{2} \nabla \log \widetilde{\mu}_t - \widetilde{b}_t \right) \widetilde{\mu}_t \right].$$

Defining $\phi(x) := x \log x$ and $\phi'(x) = \frac{\mathrm{d}}{\mathrm{d}x}\phi(x) = \log x + 1$, we can calculate

$$\partial_t \mathsf{KL}(\mu_t || \widetilde{\mu}_t) = \partial_t \int \phi\left(\frac{\mu_t}{\widetilde{\mu}_t}\right) \widetilde{\mu}_t$$

$$= \int \phi'\left(\frac{\mu_t}{\widetilde{\mu}_t}\right) \left( \partial_t \mu_t - \frac{\mu_t}{\widetilde{\mu}_t} \partial_t \widetilde{\mu}_t \right) + \int \phi\left(\frac{\mu_t}{\widetilde{\mu}_t}\right) \partial_t \widetilde{\mu}_t$$

$$= \int \phi'\left(\frac{\mu_t}{\widetilde{\mu}_t}\right) \left( \mathrm{div}\left[ \left(\frac{c(t)^2}{2}\nabla\log\mu_t - b_t\right)\mu_t \right] - \frac{\mu_t}{\widetilde{\mu}_t}\mathrm{div}\left[ \left(\frac{c(t)^2}{2}\nabla\log\widetilde{\mu}_t - \widetilde{b}_t\right)\widetilde{\mu}_t \right] \right)$$

$$\quad + \int \phi\left(\frac{\mu_t}{\widetilde{\mu}_t}\right) \mathrm{div}\left[ \left(\frac{c(t)^2}{2}\nabla\log\widetilde{\mu}_t - \widetilde{b}_t\right)\widetilde{\mu}_t \right]$$

$$= -\int \left\langle \nabla\phi'\left(\frac{\mu_t}{\widetilde{\mu}_t}\right), \frac{c(t)^2}{2}\nabla\log\mu_t - b_t \right\rangle \mu_t + \int \left\langle \nabla\left[\phi'\left(\frac{\mu_t}{\widetilde{\mu}_t}\right)\frac{\mu_t}{\widetilde{\mu}_t}\right], \frac{c(t)^2}{2}\nabla\log\widetilde{\mu}_t - \widetilde{b}_t \right\rangle \widetilde{\mu}_t$$

$$\quad - \int \left\langle \nabla\phi\left(\frac{\mu_t}{\widetilde{\mu}_t}\right), \frac{c(t)^2}{2}\nabla\log\widetilde{\mu}_t - \widetilde{b}_t \right\rangle \widetilde{\mu}_t$$

$$= -\int \left\langle \nabla\phi'\left(\frac{\mu_t}{\widetilde{\mu}_t}\right), \frac{c(t)^2}{2}\nabla\log\left(\frac{\mu_t}{\widetilde{\mu}_t}\right) - b_t + \widetilde{b}_t \right\rangle \mu_t + \int \left\langle \nabla\frac{\mu_t}{\widetilde{\mu}_t}, \frac{c(t)^2}{2}\nabla\log\widetilde{\mu}_t - \widetilde{b}_t \right\rangle \phi'\left(\frac{\mu_t}{\widetilde{\mu}_t}\right)\widetilde{\mu}_t$$

$$\quad - \int \left\langle \nabla\frac{\mu_t}{\widetilde{\mu}_t}, \frac{c(t)^2}{2}\nabla\log\widetilde{\mu}_t - \widetilde{b}_t \right\rangle \phi'\left(\frac{\mu_t}{\widetilde{\mu}_t}\right)\widetilde{\mu}_t$$

$$= -\frac{c(t)^2}{2}\int \left\| \nabla\log\left(\frac{\mu_t}{\widetilde{\mu}_t}\right) \right\|_2^2 \mu_t - \int \left\langle \nabla\log\left(\frac{\mu_t}{\widetilde{\mu}_t}\right), \widetilde{b}_t - b_t \right\rangle \mu_t$$

$$\leq -\frac{c(t)^2}{4}\int \left\| \nabla\log\left(\frac{\mu_t}{\widetilde{\mu}_t}\right) \right\|_2^2 \mu_t + \frac{1}{c(t)^2}\int \left\| \widetilde{b}_t - b_t \right\|_2^2 \mu_t$$

$$= -\frac{c(t)^2}{4}\int \left\| \nabla\log\left(\frac{\mu_t}{\widetilde{\mu}_t}\right) \right\|_2^2 \mu_t + \frac{1}{c(t)^2}\int \left\| \widetilde{b}_t - b_t \right\|_2^2 \mu_t$$

$$= -\frac{c(t)^2}{4}\mathsf{FI}(\mu_t || \widetilde{\mu}_t) + \frac{1}{c(t)^2}\int \left\| \widetilde{b}_t - b_t \right\|_2^2 \mu_t$$

where we used the fact that $-\frac{1}{2}a^2 - ab \leq -\frac{1}{4}a^2 + b^2, \forall a, b \in \mathbb{R}$ for the inequality. $\qquad\square$

Now we are ready to prove Theorem 3.1.

*Proof of Theorem 3.1.* We first consider the likelihood steps over $K$ iterations of PnP-DM. Applying Lemma 2 of [79] to the likelihood steps (9) of the non-stationary and stationary processes, we have that

$$\partial_\tau \mathsf{KL}(\pi_\tau || \nu_\tau) = -\frac{\rho^2}{2}\mathsf{FI}(\pi_\tau || \nu_\tau) \leq -\frac{\rho^2}{4}\mathsf{FI}(\pi_\tau || \nu_\tau),$$

for $\tau \in [k(t^* + 1), k(t^* + 1) + 1]$ with $k = 0, ..., K - 1$. Integrating both sides over $\tau \in [k(t^* + 1), k(t^* + 1) + 1]$, we get

$$\int_{k(t^*+1)}^{k(t^*+1)+1} \mathsf{FI}(\pi_\tau || \nu_\tau) \, \mathrm{d}\tau = \frac{4[\mathsf{KL}(\pi^X || \nu_k^X) - \mathsf{KL}(\pi^Z || \nu_k^Z)]}{\rho^2} \tag{16}$$

for $k = 0, ..., K - 1$.

Then, applying Lemma A.4 to the prior steps (12) with

$$b(\boldsymbol{x}_t, t) := u(t)\boldsymbol{x}_t - v(t)^2 \nabla \log p_t(\boldsymbol{x}_t)$$
$$\widetilde{b}(\boldsymbol{x}_t, t) := u(t)\boldsymbol{x}_t - v(t)^2 \boldsymbol{s}_t(\boldsymbol{x}_t)$$
$$c(t) := v(t)$$
$$\delta := \inf_{t \in [0, t^*]} v(t),$$

we have that

$$\partial_\tau \mathsf{KL}(\pi_\tau || \nu_\tau) \leq -\frac{v(\tau)^2}{4}\mathsf{FI}(\pi_\tau || \nu_\tau) + \frac{1}{v(\tau)^2}\int \left\| v(\tau)^2 (\boldsymbol{s}_\tau - \nabla \log p_\tau) \right\|_2^2 \pi_\tau$$

$$\leq -\frac{v(\tau)^2}{4}\mathsf{FI}(\pi_\tau || \nu_\tau) + v(\tau)^2 \int \left\| \boldsymbol{s}_\tau - \nabla \log p_\tau \right\|_2^2 \pi_\tau$$

$$\leq -\frac{\delta^2}{4}\mathsf{FI}(\pi_\tau || \nu_\tau) + v(\tau)^2 \mathbb{E}_{\pi_\tau} \left\| \boldsymbol{s}_\tau - \nabla \log p_\tau \right\|_2^2,$$

for $\tau \in [k(t^* + 1) + 1, (k + 1)(t^* + 1)]$ with $k = 0, ..., K - 1$. Integrating both sides over $\tau \in [k(t^* + 1) + 1, (k + 1)(t^* + 1)]$, we get

$$\int_{k(t^*+1)+1}^{(k+1)(t^*+1)} \mathsf{FI}(\pi_\tau || \nu_\tau) \, \mathrm{d}\tau \leq \frac{4[\mathsf{KL}(\pi^Z || \nu_k^Z) - \mathsf{KL}(\pi^X || \nu_{k+1}^X)]}{\delta^2} + \frac{4\epsilon_{\text{score}}}{\delta^2} \tag{17}$$

where

$$\epsilon_{\text{score}} := \int_{k(t^*+1)+1}^{(k+1)(t^*+1)} v(\tau)^2 \mathbb{E}_{\pi_\tau} \left\| \boldsymbol{s}_\tau - \nabla \log p_\tau \right\|_2^2 \mathrm{d}\tau = \int_1^{t^*+1} v(\tau)^2 \mathbb{E}_{\pi_\tau} \left\| \boldsymbol{s}_\tau - \nabla \log p_\tau \right\|_2^2 \mathrm{d}\tau.$$

Finally, combining (16) and (17) for $k = 0, ..., K - 1$, we obtain

$$\int_0^T \mathsf{FI}(\pi_\tau || \nu_\tau) \, \mathrm{d}\tau \leq \frac{4[\mathsf{KL}(\pi^X || \nu_0^X) - \mathsf{KL}(\pi^X || \nu_K^X)]}{\min(\rho, \delta)^2} + \frac{4K\epsilon_{\text{score}}}{\delta^2}$$

$$\leq \frac{4\mathsf{KL}(\pi^X || \nu_0^X)}{\min(\rho, \delta)^2} + \frac{4K\epsilon_{\text{score}}}{\delta^2}.$$

The proof is concluded by dividing $T = K(t^* + 1)$ on both sides. $\qquad\square$

## A.3 Discussion

To facilitate the discussion, we first present the following proposition.

**Proposition A.5.** *Define a weighting function* $\lambda(\tau)$ *over* $\tau \in [0, T]$ *such that for* $k = 0, ..., K - 1$,

$$\lambda(\tau) = \begin{cases} \rho^2 & \text{if } \tau \in [k(t^* + 1), k(t^* + 1) + 1], \\ v(\tau)^2 & \text{if } \tau \in [k(t^* + 1) + 1, (k + 1)(t^* + 1)]. \end{cases}$$

*Then, under the same settings of Theorem 3.1, we have*

$$\frac{1}{T} \int_0^T \lambda(\tau) \mathsf{FI}\left(\pi_\tau || \nu_\tau\right) \mathrm{d}\tau = \frac{4\mathsf{KL}(\pi^X || \nu_0^X)}{K(t^*+1)} + \frac{4\epsilon_{score}}{t^*+1} \tag{18}$$

*where $\epsilon_{score} := \int_1^{t^*+1} v(\tau)^2 \mathbb{E}_{\pi_\tau} \|s_\tau - \nabla \log p_\tau\|_2^2 \mathrm{d}\tau$.*

*Proof.* With the definition of $\lambda(\tau)$, we can apply Lemma 2 of [79] to the likelihood steps and obtain

$$\int_{k(t^*+1)}^{k(t^*+1)+1} \lambda(\tau) \mathsf{FI}\left(\pi_\tau || \nu_\tau\right) \mathrm{d}\tau = 4[\mathsf{KL}(\pi^X || \nu_k^X) - \mathsf{KL}(\pi^Z || \nu_k^Z)] \tag{19}$$

for $k = 0, ..., K-1$. Similarly, we can apply Lemma A.4 to the prior steps and obtain

$$\int_{k(t^*+1)+1}^{(k+1)(t^*+1)} \lambda(\tau) \mathsf{FI}\left(\pi_\tau || \nu_\tau\right) \mathrm{d}\tau \leq 4[\mathsf{KL}(\pi^Z || \nu_k^Z) - \mathsf{KL}(\pi^X || \nu_{k+1}^X)] + 4\epsilon_{\text{score}} \tag{20}$$

where

$$\epsilon_{\text{score}} := \int_{k(t^*+1)+1}^{(k+1)(t^*+1)} v(\tau)^2 \mathbb{E}_{\pi_\tau} \|s_\tau - \nabla \log p_\tau\|_2^2 \mathrm{d}\tau = \int_1^{t^*+1} v(\tau)^2 \mathbb{E}_{\pi_\tau} \|s_\tau - \nabla \log p_\tau\|_2^2 \mathrm{d}\tau.$$

Together, for $\tau \in [0, T]$. We can then get (18) by combining (19) and (20) for $k = 0, ..., K-1$ and dividing by $T := K(t^*+1)$. $\qquad\square$

Unlike Theorem 3.1, this proposition calculates the weighted average of the Fisher information along the two processes with the weighting function $\lambda(\tau)$. The bound in Theorem 3.1 on the unweighted average of Fisher information can be obtained by further lower-bounding the left hand side of (18) using the infimum of $\lambda(\tau)$ over $\tau \in [0, T]$. Given this observation, we can see the role of $\delta$ in Theorem 3.1. With a strictly positive $\delta$, the weighting function $\lambda(\tau)$ is always strictly positive, so the (unweighted) average Fisher information must converge to 0. This is precisely the case for the VP- and VE-SDE [63]. On the other hand, if $\delta = 0$, the Fisher information $\mathsf{FI}\left(\pi_\tau || \nu_\tau\right)$ may be increasingly large as $\lambda(\tau)$ gets closer to 0. For iDDPM and EDM, this could happen near $t = 0$ in the reverse diffusion at $v(0) = 0$ for these diffusion processes. Nevertheless, we can instead consider a slightly adjusted diffusion coefficient $\tilde{v}(t) := v(t) + \epsilon$ with $\epsilon > 0$. Using the relation between scores and diffusions $\mathrm{div}(p\nabla \log p) = \Delta p$, we get the following reverse SDE which has the same law as (7) at each $t$:

$$\mathrm{d}x_t = \left[\frac{\dot{s}(t)}{s(t)} x_t + \left(\frac{\epsilon^2}{2} - 2s(t)^2 \dot{\sigma}(t)\sigma(t)\right) \nabla \log p\left(\frac{x_t}{s(t)}; \sigma(t)\right)\right] \mathrm{d}t + \left(s(t)\sqrt{2\dot{\sigma}(t)\sigma(t)} + \epsilon\right) \mathrm{d}\bar{w}_t.$$

In this case, $\tilde{v}(t) = s(t)\sqrt{2\dot{\sigma}(t)\sigma(t)} + \epsilon$ is strictly positive, so the convergence on the unweighted average Fisher information is also guaranteed.

## B   Inverse problem setup

**Data usage**   We list the data we have used for our experiments:

- For the synthetic prior experiment, we took images from the CelebA dataset [45], turned them into grayscale, rescaled them to $[-1, 1]$, and resized them to $32 \times 32$ pixels for efficient computation. We then found the empirical mean and covariance of the images to construct the Gaussian image prior. The test image was randomly drawn from this Gaussian prior.

- For the benchmark experiments, we used the first 100 images (index 00000 to 00099) in the FFHQ dataset [38]. For all linear inverse problems, the test images were in RGB and normalized to range $[-1, 1]$. For all nonlinear problems, the test images were in grayscale and normalized to range $[0, 1]$.

- For the black hole experiments, we used the simulated data used in [64] and the publicly available EHT 2017 data[1] that was used to produce the first image of the M87 black hole.

---

[1] https://eventhorizontelescope.org/blog/public-data-release-event-horizon-telescope-2017-observations

**Gaussian and motion deblur**   The forward model is defined as

$$\boldsymbol{y} \sim \mathcal{N}(\boldsymbol{Bx}, \sigma_{\boldsymbol{y}}^2 \boldsymbol{I})$$

where $\boldsymbol{B} \in \mathbb{R}^{n \times n}$ is a circulant matrix that effectively implements a convolution with kernel $\boldsymbol{k}$ under the circular boundary condition. For the Gaussian deblurring problem, we fixed the kernel $\boldsymbol{k}$ as a Gaussian kernel with standard deviation $3.0$ and size $61 \times 61$. For the motion deblurring problem, we randomly generated the kernel $\boldsymbol{k}$ for each test image using the code[2] with intensity of $0.5$ and size $61 \times 61$. For fair comparison, the blur kernel for each test image was set the same for all compared methods.

**Super-resolution**   The forward model is defined as

$$\boldsymbol{y} \sim \mathcal{N}(\boldsymbol{P}_f \boldsymbol{x}, \sigma_{\boldsymbol{y}}^2 \boldsymbol{I})$$

where $\boldsymbol{P}_f \in \mathbb{R}^{\frac{n}{f} \times n}$ is a matrix that implements a block averaging filter to downscale the images by a factor of $f$. Specifically, we set $f = 4$ and used the SVD implementation from the code[3] of [39].

**Coded diffraction patterns (CDP)**   CDP is a measurement model originally proposed in [10]. The target $\boldsymbol{x}$ is illuminated by a coherent source and modulated by a phase mask $\boldsymbol{D}$. The light field then undergoes the far-field Fraunhofer diffraction and is measured by a standard camera. Mathematically, the forward model of CDP is defined as

$$\boldsymbol{y} \sim \mathcal{N}(|\boldsymbol{FDx}|, \sigma_{\boldsymbol{y}}^2 \boldsymbol{I})$$

where $\boldsymbol{F}$ denotes the 2D Fourier transform. We followed [75] to set $\boldsymbol{D}$ as a diagonal matrix with entries drawn randomly from the complex unit circle.

**Fourier phase retrieval**   We adopted a similar setting as [18]. In particular, the forward model is defined as

$$\boldsymbol{y} \sim \mathcal{N}(|\boldsymbol{FPx}|, \sigma_{\boldsymbol{y}}^2 \boldsymbol{I}),$$

where $\boldsymbol{P}$ denotes the oversampling matrix that effectively pads $\boldsymbol{x}$ in 2D matrix form with zeros. We considered a $4\times$ oversampling ratio for grayscale images of size $256 \times 256$, so $\boldsymbol{Px}$ has a size of $512 \times 512$.

**Black hole imaging**   We adopted the same BHI setup as in [64, 66]. The relationship between the black hole image and each interferometric measurement, or so-called *visibility*, is given by

$$V_{a,b}^t = g_a^t g_b^t \cdot e^{-i(\phi_a^t - \phi_b^t)} \cdot \boldsymbol{F}_{a,b}^t(\boldsymbol{x}) + \eta_{a,b} \in \mathbb{C}, \tag{21}$$

where $a$ and $b$ denote a pair of telescopes, $t$ represents the time of measurement acquisition, $i$ is the imaginary unit, and $\boldsymbol{F}_{a,b}^t(\boldsymbol{x})$ is the Fourier component of the image $\boldsymbol{x}$ corresponding to the baseline between telescopes $a$ and $b$ at time $t$. In practice, there are three main sources of noise in (21): gain error $g_a$ and $g_b$ at the telescopes, phase error $\phi_a^t$ and $\phi_b^t$, and baseline-based additive white Gaussian noise $\eta_{a,b}$. The gain and phase errors stem from atmospheric turbulence and instrument miscalibration and often cannot be ignored. To correct for these two errors, multiple noisy visibilities can be combined into data products that are invariant to these errors, which are called *closure phase* and *log closure amplitude* measurements [12]

$$\boldsymbol{y}_{t,(a,b,c)}^{\mathsf{cph}} = \angle(V_{a,b}V_{b,c}V_{a,c}) := \mathcal{A}_{t,(a,b,c)}^{\mathsf{cph}}(\boldsymbol{x}),$$

$$\boldsymbol{y}_{t,(a,b,c,d)}^{\mathsf{logcamp}} = \log\left(\frac{|V_{a,b}^t||V_{c,d}^t|}{|V_{a,c}^t||V_{b,d}^t|}\right) := \mathcal{A}_{t,(a,b,c,d)}^{\mathsf{logcamp}}(\boldsymbol{x}),$$

where $\angle$ computes the angle of a complex number. Given a total of $M$ telescopes, there are in total $\frac{(M-1)(M-2)}{2}$ closure phase and $\frac{M(M-3)}{2}$ log closure amplitude measurements at time $t$, after eliminating repetitive measurements. In our experiments, we used a 9-telescope array ($M = 9$) from

---

[2]https://github.com/LeviBorodenko/motionblur (license unknown)
[3]https://github.com/bahjat-kawar/ddrm (MIT license)

the Event Horizon Telescope (EHT) and constructed the data likelihood term based on these nonlinear closure quantities. Additionally, because the closure quantities do not constrain the total flux (i.e. summation of the pixel values) of the underlying black hole image, we added a constraint on the total flux in the likelihood term. The overall potential function of the likelihood is given by

$$f(\boldsymbol{x}; \boldsymbol{y}) = \sum_{t,\mathsf{c}} \frac{\|\mathcal{A}_{t,\mathsf{c}}^{\mathsf{cph}}(\boldsymbol{x}) - \boldsymbol{y}_{t,\mathsf{c}}^{\mathsf{cph}}\|_2^2}{2\sigma_{\mathsf{cph}}^2} + \sum_{t,\mathsf{d}} \frac{\|\mathcal{A}_{t,\mathsf{d}}^{\mathsf{logcamp}}(\boldsymbol{x}) - \boldsymbol{y}_{t,\mathsf{d}}^{\mathsf{logcamp}}\|_2^2}{2\sigma_{\mathsf{logcamp}}^2} + \frac{\left\|\sum_i \boldsymbol{x}_i - \boldsymbol{y}^{\mathsf{flux}}\right\|_2^2}{2\sigma_{\mathsf{flux}}^2}. \quad (22)$$

In this equation, $\boldsymbol{y}^{\mathsf{flux}}$ is the total flux of the underlying black hole, which can be accurately measured. We use $\boldsymbol{y} := (\boldsymbol{y}^{\mathsf{cph}}, \boldsymbol{y}^{\mathsf{logcamp}}, \boldsymbol{y}^{\mathsf{flux}})$ to denote all the measurements and c, d as the indices for the closure phase and log closure amplitude measurements. Parameters $\sigma_{\mathsf{cph}}$, $\sigma_{\mathsf{logcamp}}$ were given by the telescope system and $\sigma_{\mathsf{flux}}$ was set to $\sqrt{2}$ in our experiments to constrain the total flux. The data mismatch metric reported in Figure 8 is defined as the sum of the reduced $\chi^2$ values for the closure phase and log closure amplitude measurements, which are calculated using the `ehtim.obsdata.Obsdata.chisq` function of the `ehtim` package[4]. Both $\chi^2$ values should ideally be around 1 for data with high signal-to-noise ratio (SNR). Therefore, a data mismatch value around 2 to 3 is considered as fitting the measurements well.

## C Technical details of PnP-DM

### C.1 Likelihood step

**Linear forward model and Gaussian noise** As we showed in the main paper, in case of linear forward models and Gaussian noise, the likelihood step is

$$\pi^{Z|X=\boldsymbol{x}} = \mathcal{N}(\boldsymbol{m}(\boldsymbol{x}), \boldsymbol{\Lambda}^{-1})$$

where $\boldsymbol{\Lambda} := \boldsymbol{A}^T \boldsymbol{\Sigma}^{-1} \boldsymbol{A} + \frac{1}{\rho^2} \boldsymbol{I}$ and $\boldsymbol{m}(\boldsymbol{x}) := \boldsymbol{\Lambda}^{-1}(\boldsymbol{A}^T \boldsymbol{\Sigma}^{-1} \boldsymbol{y} + \frac{1}{\rho^2} \boldsymbol{x})$. The bottleneck here is that both the mean and the covariance involve the matrix inverse $\boldsymbol{\Lambda}^{-1}$, which can be prohibitive to compute directly for high-dimensional problems. Nevertheless, the computational cost can be significantly alleviated when the noise is i.i.d. Gaussian, i.e. $\boldsymbol{\Sigma} = \sigma_{\boldsymbol{y}}^2 \boldsymbol{I}$, and $\boldsymbol{A}$ can be efficiently decomposed. For example, if one can efficiently calculate the SVD of the forward model $\boldsymbol{A}$, i.e. $\boldsymbol{A} = \boldsymbol{U}\boldsymbol{S}\boldsymbol{V}^T$, one can find the Cholesky decomposition of $\boldsymbol{\Lambda}^{-1}$ as

$$\boldsymbol{\Lambda}^{-1} = \boldsymbol{L}\boldsymbol{L}^T \quad \text{where} \quad \boldsymbol{L} := \boldsymbol{V}\left(\frac{1}{\sigma^2}\boldsymbol{S}^2 + \frac{1}{\rho^2}\boldsymbol{I}\right)^{-1/2}.$$

Since $\boldsymbol{S}$ is a diagonal matrix, the second term can be calculated with only $O(n)$ complexity. Then, leveraging the property of multivariate Gaussian distribution, we can sample $\boldsymbol{\eta} \sim \mathcal{N}(\boldsymbol{0}, \boldsymbol{I})$ and calculate $\boldsymbol{z} = \boldsymbol{m}(\boldsymbol{x}) + \boldsymbol{L}\boldsymbol{\eta}$ as a sample that exactly follows the target Gaussian distribution $\mathcal{N}(\boldsymbol{m}(\boldsymbol{x}), \boldsymbol{\Lambda}^{-1})$. An analogous derivation with Fourier transform can be done when $\boldsymbol{A}$ is a circulant convolution matrix.

**Nonlinear forward model** We provide the pseudocode of the LMC algorithm for sampling the likelihood step with general differentiable forward models in Algorithm 2.

---

**Algorithm 2** Langevin Monte Carlo for the likelihood step under general $\mathcal{A}$

---

**Input:** state $\boldsymbol{x}$, coupling strength $\rho > 0$, likelihood potential $f(\,\cdot\,; \boldsymbol{y})$ with measurements $\boldsymbol{y}$
**Hyperparameter:** step size $\gamma > 0$, number of iterations $J > 0$
1: $\boldsymbol{u}_0 \leftarrow \boldsymbol{x}$
2: **for** $j = 0, \cdots, J-1$ **do**
3: $\quad \boldsymbol{u}_{j+1} \leftarrow \boldsymbol{u}_j - \gamma \nabla f(\boldsymbol{u}_j; \boldsymbol{y}) - \frac{\gamma}{\rho^2}(\boldsymbol{u}_j - \boldsymbol{x}) + \sqrt{2\gamma}\boldsymbol{\epsilon}_j \quad$ where $\quad \boldsymbol{\epsilon}_j \sim \mathcal{N}(\boldsymbol{0}, \boldsymbol{I})$
4: **end for**
5: **return** $\boldsymbol{u}_J$

---

Table 3 summarizes the hyperparameters we used for solving the nonlinear inverse problems considered in this work.

---

[4]https://github.com/achael/eht-imaging (GPL-3.0 license)

Table 3: List of hyperparameters for the likelihood step of PnP-DM

| Inverse problem | Step size ($\gamma$) | Number of iterations ($J$) |
|---|---|---|
| Coded diffraction patterns | 1.0e-3 | 100 |
| Fourier phase retrieval | 1.0e-4 | 100 |
| Black hole imaging | 1.0e-5 | 200 |

## C.2 Prior step

**The EDM framework** We formally introduce the EDM formulation [37] using our notations. The forward diffusion process is defined as the following linear Itô SDE

$$\mathrm{d}\boldsymbol{x}_t = u(t)\boldsymbol{x}_t\mathrm{d}t + v(t)\mathrm{d}\boldsymbol{w}_t, \tag{23}$$

where $u(t) : \mathbb{R} \to \mathbb{R}$, $v(t) : \mathbb{R} \to \mathbb{R}$ are the drift and diffusion coefficients. The generative process is the time-reversed version of (23). According to [2], it is another Itô SDE of the form

$$\mathrm{d}\boldsymbol{x}_t = \left[u(t)\boldsymbol{x}_t - v(t)^2\nabla_{\boldsymbol{x}_t}\log p_t(\boldsymbol{x}_t)\right]\mathrm{d}t + v(t)\mathrm{d}\bar{\boldsymbol{w}}_t, \tag{24}$$

where $p_t(\boldsymbol{x}_t)$ is the marginal distribution of $\boldsymbol{x}_t$. There also exists a reverse probability flow ODE

$$\mathrm{d}\boldsymbol{x}_t = \left[u(t)\boldsymbol{x}_t - \frac{1}{2}v(t)^2\nabla_{\boldsymbol{x}_t}\log p_t(\boldsymbol{x}_t)\right]\mathrm{d}t, \tag{25}$$

which shares the same marginal distributions as (24). Based on (23), we have

$$p(\boldsymbol{x}_t|\boldsymbol{x}_0) = \mathcal{N}(s(t)\boldsymbol{x}_0, s(t)^2\sigma(t)^2\boldsymbol{I})$$

where $s(t) := \exp\left(\int_0^t u(\xi)\mathrm{d}\xi\right)$ and $\sigma(t) := \sqrt{\int_0^t \frac{v(\xi)^2}{s(\xi)^2}\,\mathrm{d}\xi}$. We also have $\boldsymbol{x}_t/s(t) \sim p(\boldsymbol{x}; \sigma(t))$ where $p(\boldsymbol{x}; \sigma(t))$ is the distribution obtained by adding i.i.d. Gaussian noise of standard deviation $\sigma(t))$ to the prior data. The idea of the EDM formulation is to write the reverse diffuison directly in terms of the scaling and noise level of $\boldsymbol{x}_t$ with respect to $\boldsymbol{x}_0$, which are more important than the drift and diffusion coefficients. With the relations between $u(t)$, $v(t)$, $p_t$ and $s(t)$, $\sigma(t)$, $p(\cdot; \sigma(t))$, we can rewrite (24) and (25) as

$$\mathrm{d}\boldsymbol{x}_t = \left[\frac{\dot{s}(t)}{s(t)}\boldsymbol{x}_t - 2s(t)^2\dot{\sigma}(t)\sigma(t)\nabla_{\boldsymbol{x}_t}\log p\left(\frac{\boldsymbol{x}_t}{s(t)}; \sigma(t)\right)\right]\mathrm{d}t + s(t)\sqrt{2\dot{\sigma}(t)\sigma(t)}\mathrm{d}\bar{\boldsymbol{w}}_t \tag{26}$$

and

$$\mathrm{d}\boldsymbol{x}_t = \left[\frac{\dot{s}(t)}{s(t)}\boldsymbol{x}_t - s(t)^2\dot{\sigma}(t)\sigma(t)\nabla_{\boldsymbol{x}_t}\log p\left(\frac{\boldsymbol{x}_t}{s(t)}; \sigma(t)\right)\right]\mathrm{d}t. \tag{27}$$

Note that (26) is precisely the SDE (7) we considered for the prior step. Finally, due to the Tweedie's formula [24], we can approximate $\nabla_{\boldsymbol{x}_t}\log p\left(\cdot; \sigma(t)\right)$ by a denoiser $[D_\theta(\cdot; \sigma(t)) - \cdot]/\sigma(t)^2$ trained to minimize the $\ell_2$ error of a denoising objective. Substituting the score function by the approximation with the denoiser and using the chain rule, we can further rewrite (26) and (27) as

$$\mathrm{d}\boldsymbol{x}_t = \left[\left(\frac{2\dot{\sigma}(t)}{\sigma(t)} + \frac{\dot{s}(t)}{s(t)}\right)\boldsymbol{x}_t - \frac{2\dot{\sigma}(t)s(t)}{\sigma(t)}D_\theta\left(\frac{\boldsymbol{x}_t}{s(t)}; \sigma(t)\right)\right]\mathrm{d}t + s(t)\sqrt{2\dot{\sigma}(t)\sigma(t)}\mathrm{d}\bar{\boldsymbol{w}}_t \tag{28}$$

and

$$\mathrm{d}\boldsymbol{x}_t = \left[\left(\frac{\dot{\sigma}(t)}{\sigma(t)} + \frac{\dot{s}(t)}{s(t)}\right)\boldsymbol{x}_t - \frac{\dot{\sigma}(t)s(t)}{\sigma(t)}D_\theta\left(\frac{\boldsymbol{x}_t}{s(t)}; \sigma(t)\right)\right]\mathrm{d}t. \tag{29}$$

**Pseudocode** We provide the pseudocode for our prior step in Algorithm 3. Note that the update rule is precisely the Euler discretization of (28) and (29). The discretization time steps $\{t_i\}_{i=0}^N$, scaling schedule $s(\cdot)$, and noise schedule $\sigma(\cdot)$, are kept the same as in Table 1 of [37]. For all experiments, we set the total number of time steps to 100, i.e. $N = 100$. We note that this does not imply that each prior step has a number of function evaluations (NFE) equal to 100. Since $\rho$ is to a small value as the algorithm runs, the number of steps in later iterations of the algorithm are much fewer than 100. The prior step is similar to the image synthesis process in SDEdit [50] that starts from the middle of the reverse diffusion process. We used the `pf-ODE` solver for the CDP problem and the `SDE` solver for all other problems. Part of our code implementation is based on the repository[5].

---

[5]https://github.com/NVlabs/edm/tree/main (Creative Commons Attribution-NonCommercial-ShareAlike 4.0 International Public license)

**Algorithm 3** EDM for the prior step (Bayesian denoising with noise level $\rho$)

---

**Input:** noisy image $\boldsymbol{z} \in \mathbb{R}^n$, assumed noise level $\rho > 0$, pretrained model $D_\theta(\,\cdot\,;\,\cdot\,)$ that approximates $\nabla \log p(\boldsymbol{x};\sigma)$ with $(D_\theta(\boldsymbol{x};\sigma) - \boldsymbol{x})/\sigma^2$

**Hyperparameter:** discretization time steps $\{t_i\}_{i=0}^N$ (monotonically decreasing to $t_N = 0$), scaling schedule $s(\cdot)$, noise schedule $\sigma(\cdot)$, solver (SDE or pf-ODE)

1: $i^* \leftarrow$ smallest $i$ such that $\sigma(t_i) \le \rho$      ▷ Find the starting point of the reverse diffusion
2: $\boldsymbol{v}_{i^*} \leftarrow s(t_{i^*})\boldsymbol{z}$      ▷ Initialize at time $t_{i^*}$
3: **for** $i = i^*, \cdots, N-1$ **do**
4:      $\lambda \leftarrow 2$ **if** solver is SDE **else** 1
5:      $\boldsymbol{d}_i \leftarrow \left( \frac{\lambda \dot{\sigma}(t_i)}{\sigma(t_i)} + \frac{\dot{s}(t_i)}{s(t_i)} \right) \boldsymbol{v}_i - \frac{\lambda \dot{\sigma}(t_i) s(t_i)}{\sigma(t_i)} D_\theta \left( \frac{\boldsymbol{v}_i}{s(t_i)}; \sigma(t_i) \right)$
6:      $\boldsymbol{v}_{i+1} \leftarrow \boldsymbol{v}_i + (t_{i+1} - t_i)\boldsymbol{d}_i$      ▷ Drift
7:      **if** $i \neq N-1$ **and** solver is SDE **then**
8:          $\boldsymbol{v}_{i+1} \leftarrow \boldsymbol{v}_{i+1} + s(t)\sqrt{2\dot{\sigma}(t)\sigma(t)(t_i - t_{i+1})}\boldsymbol{\epsilon}_i$    where    $\boldsymbol{\epsilon}_i \sim \mathcal{N}(\boldsymbol{0}, \boldsymbol{I})$      ▷ Diffusion
9:      **end if**
10: **end for**
11: **return** $\boldsymbol{v}_N$

---

**Model checkpoint** For experiments with FFHQ color images, we used the pretrained checkpoint from [16] available at the repository[6]. For experiments with synthetic data, FFHQ grayscale images, and black hole images, we trained our own models using the same repository. The model network is based on the U-Net architecture in [52] with BigGAN [9] residual blocks, multi-resolution attention, and multi-head attention with fixed channels per head. See the appendix of [16] for architecture details. Specifically, we changed the input and output channels to 1 and 2, respectively, to accommodate grayscale inputs, and reduced the number of down-pooling and up-pooling levels in the U-Net for smaller images. We trained all models until convergence using an exponential moving average (EMA) rate of 0.9999, 32-bit precision, and the AdamW optimizer [46]. Here is a list of training data we used for each model:

- For the Gaussian prior model, we randomly generated images from the constructed Gaussian prior distribution.

- For the FFHQ grayscale model, we used the images with index 01000 to 69999 in the FFHQ dataset.

- For the black hole model, we used 3068 simulated black hole images from the GRMHD simulation, which stands for *general relativistic magnetohydrodynamic* simulation [22]. See Figure 9 for some example training images. We applied data augmentation with random flipping and resizing, so that the flux spin rotation and the ring diameter vary from sample to sample.

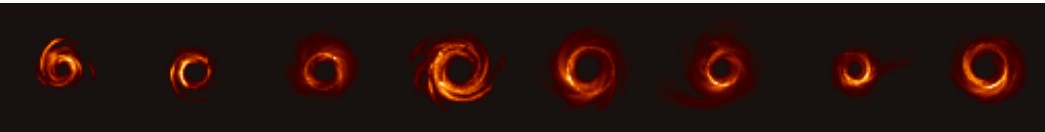

Figure 9: Example images from the dataset for training the black hole diffusion model prior.

**Preconditioning** Since the checkpoints we used are all trained based on the DDPM (or VP-SDE) formulation [33], we converted them to the denoiser $D_\theta$ under the EDM formulation via the VP preconditioning [37]. Specifically, if we denote the pretrained model as $F_\theta(\,\cdot\,;\,\cdot\,)$, the model we used for Algorithm 3 is

$$D_\theta(\boldsymbol{x};\sigma) := c_{\text{skip}}(\sigma)\boldsymbol{x} + c_{\text{out}}(\sigma)F_\theta\left(c_{\text{in}}(\sigma)\boldsymbol{x}; c_{\text{noise}}(\sigma)\right), \tag{30}$$

where $c_{\text{skip}}(\sigma) = 1$, $c_{\text{out}}(\sigma) = -\sigma$, $c_{\text{in}}(\sigma) = 1/\sqrt{\sigma^2 + 1}$, and $c_{\text{noise}}(\sigma) = 999\sigma_{\text{VP}}^{-1}(\sigma)$. Here $\sigma_{\text{VP}}^{-1}(\cdot)$ is the inverse of the VP-SDE noise schedule defined as $\sigma_{\text{VP}}(t) := \sqrt{e^{\frac{1}{2}\beta_{\text{d}}t^2 + \beta_{\min}t} - 1}$ with $\beta_{\text{d}} = 19.9$ and $\beta_{\min} = 0.1$. This adaption allows us to make a fair comparison with other DM-based methods using the same pretrained models. One can also incorporate DMs trained with other formulations into PnP-DM by properly setting the preconditioning parameters.

---

[6]https://github.com/jychoi118/P2-weighting (MIT license)

**Connection to `DDS-DDPM` in [78]** A concurrent work [78] introduced a rigorous implementation of the prior step, called `DDS-DDPM`, by converting the DDPM [33] (or VP-SDE [63]) sampler into a reverse diffusion based on the VE-SDE [63]. The diffusion process after the conversion can be used to solve (6) rigorously by properly choosing the starting point. In fact, our formulation admits `DDS-DDPM` as a special case with the VP preconditioning and reverse diffusion based on the VE-SDE. Here we explicitly show this connection. For the VE-SDE, we have $s_{\mathsf{VE}}(t) = 1$, $\sigma_{\mathsf{VE}}(t) = \sqrt{t}$, $u_{\mathsf{VE}}(t) = 0$, and $v_{\mathsf{VE}}(t) = 1$. So (28) becomes

$$\mathrm{d}\boldsymbol{x}_t = \left[\frac{1}{t}\boldsymbol{x}_t - \frac{1}{t}D_\theta\left(\boldsymbol{x}_t; \sqrt{t}\right)\right]\mathrm{d}t + \mathrm{d}\bar{\boldsymbol{w}}_t \tag{31}$$

Applying the VP preconditioning (30) to (31), we obtain

$$\mathrm{d}\boldsymbol{x}_t = \left[\frac{1}{\sqrt{t}}F_\theta\left(\frac{\boldsymbol{x}_t}{\sqrt{t+1}}; 999\sigma_{\mathsf{VP}}^{-1}(\sqrt{t})\right)\right]\mathrm{d}t + \mathrm{d}\bar{\boldsymbol{w}}_t. \tag{32}$$

We can then rescale the time range from $[0,1]$ to $[0,1000]$, discretize (32) backward in time over the time steps $\{\tau_t\}$ from [78], and apply the exponential integrator [83] to the drift term, resulting in the following update rule:

$$\widehat{\boldsymbol{x}}_{t-1} = \widehat{\boldsymbol{x}}_t - 2(\sqrt{\tau_t} - \sqrt{\tau_{t-1}})F_\theta\left(\frac{\widehat{\boldsymbol{x}}_t}{\sqrt{\tau_t + 1}}; \sigma_{\mathsf{VP}}^{-1}(\sqrt{\tau_t})\right) + \sqrt{\tau_t - \tau_{t-1}}\boldsymbol{\epsilon} \quad \text{where } \boldsymbol{\epsilon} \sim \mathcal{N}(\boldsymbol{0}, \boldsymbol{I})$$

Based on the definition $\tau_t := \bar{\alpha}_t^{-1} - 1 = \sigma_{\mathsf{VP}}(t)^2$ in `DDS-DDPM`, we get

$$\widehat{\boldsymbol{x}}_{t-1} = \widehat{\boldsymbol{x}}_t - 2(\sqrt{\tau_t} - \sqrt{\tau_{t-1}})F_\theta\left(\sqrt{\bar{\alpha}_t}\widehat{\boldsymbol{x}}_t; t\right) + \sqrt{\tau_t - \tau_{t-1}}\boldsymbol{\epsilon} \quad \text{where } \boldsymbol{\epsilon} \sim \mathcal{N}(\boldsymbol{0}, \boldsymbol{I}) \tag{33}$$

This is exactly the update rule of `DDS-DDPM` with $F_\theta(\cdot; t)$ denoting the noise estimate $\widehat{\epsilon}_t(\cdot)$ of DDPM. One can also verify that the initialization in `DDS-DDPM` is equivalent to ours by checking that $\bar{\alpha}_t \geq \frac{1}{\eta^2+1}$ is equivalent to $\tau_t = \sigma_{\mathsf{VP}}(t)^2 \leq \eta^2$ where $\eta \equiv \rho$ is the assumed noise level in (6). As one can see, `DDS-DDPM` is equivalent to our prior step by choosing the VP-preconditioning, VE reverse diffusion, and a particular integration scheme. In fact, our prior step allows for more general definitions of diffusion processes and includes both the ODE and SDE solvers.

## C.3 Others

**Annealing schedule for $\rho$** In this work, we considered an exponential annealing schedule for the coupling strength $\rho$. We note that the schedule can be more general than exponential decay and we leave the investigation of other decay schedules in future work. Specifically, we specified a starting level $\rho_0$, decay rate $\alpha$, and a minimum value $\rho_{\min}$. Then we set

$$\rho_k = \max(\alpha^k \rho_0, \rho_{\min})$$

for $k = 0, \cdots, K-1$. Table 4 summarizes the annealing hyperparameters that we used for all the inverse problems considered in this work.

Table 4: List of hyperparameters for the annealing schedule of $\rho$ in PnP-DM

| Inverse problem | Starting level ($\rho_0$) | Minimum level ($\rho_{\min}$) | Decay rate ($\alpha$) |
|---|---|---|---|
| Synthetic prior experiments | 0.03 | 0.03 | 1 |
| Gaussian deblur | 10 | 0.3 | 0.9 |
| Motion deblur | 10 | 0.3 | 0.9 |
| Super-resolution | 10 | 0.3 | 0.9 |
| Coded diffraction patterns | 10 | 0.1 | 0.9 |
| Fourier phase retrieval | 10 | 0.1 | 0.9 |
| Black hole imaging | 10 | 0.02 | 0.93 |

**Initialization** For the linear inverse problems, we used the zero initialization, i.e. $\boldsymbol{x}^{(0)} = \boldsymbol{0} \in \mathbb{R}^n$. For the CDP and Fourier phase retrieval problems, we used the Gaussian initialization, i.e. $\boldsymbol{x}^{(0)} \sim \mathcal{N}(\boldsymbol{0}, \boldsymbol{I})$. For black hole imaging experiments, we used the uniform random initialization between 0 and 1 for each pixel. We found that PnP-DM, as an MCMC algorithm, is insensitive to the initialization. Except for the black hole experiments where we found the negative values would cause problems, any reasonable initialization would lead to comparable results. This observation corroborates our convergence result, which holds for any initialization $\nu_0^X$.

**Number of iterations**   We ran 500 iterations for the synthetic prior experiments, 200 iterations for the black hole experiments, and 100 for all other experiments. The numbers were chosen so that the algorithm was fully converged.

**Sample collection**   To collect multiple samples using our method, there are two main approaches: (1) Run a single Markov chain and collect samples after a certain number of iterations, known as the burn-in period, to ensure the chain has converged. (2) Run several independent Markov chains and collect one sample from each chain after convergence. The first approach is more efficient, but the collected samples are not entirely independent and thus may have a small effective sample size. The second approach ensures all samples are fully independent but takes longer to run. In our experiments, we used the first approach for all tests involving $256 \times 256$ images to enhance efficiency. Specifically, we set the burn-in period to 40 iterations and collected 20 random samples from the remaining 60 iterations (one every 3 iterations). For other experiments, due to the smaller image sizes, we employed the second approach to obtain fully independent samples.

**Compute**   All experiments were performed on NVIDIA RTX A6000 and A100 GPUs. The runtime per image depends on several factors, such as the choice of GPU, the total number of iterations and the coupling strength schedule $\{\rho_k\}$ (as it takes more network evaluations for larger $\rho$ for our EDM-based denoiser). In our actual experiments, we ran each image for at least 100 iterations to ensure convergence, which took around 1 minute for a single Markov chain. Here we present a comparison of computational efficiency with the major baselines on a linear super-resolution and a nonlinear coded diffraction patterns problem in Table 5. The clock time in seconds and number of function evaluations (NFE) are calculated for each method to measure its computational efficiency. All hyperparameters are kept the same for each method as those used for Table 1 and Table 2 in the manuscript. As expected, DM-based approaches (DDRM & DPS) generally yield shorter runtimes due to their lower NFEs. Nevertheless, our PnP-DM method significantly outperforms these methods while achieving comparable runtimes with DPS ($\approx 1.5\times$), despite its larger NFEs ($\approx 3\times$). This is primarily due to two factors: 1) PnP-DM avoids running the full diffusion process by adapting the starting noise level to $\rho_k$ at each iteration, and 2) the runtime is further reduced by using an annealing schedule of $\rho_k$. We also note that the runtime reported for DDRM and DPS below is the time it takes to generate one sample. For the linear inverse problem experiments, where we generated 20 samples for each sampling method, PnP-DM was faster than DPS because we took 20 samples that PnP-DM generated along one Markov chain of batch size 1 (hence same runtime as below, around 50 seconds) but DPS requires running a diffusion process with batch size 20, which was significantly slower (around 330 seconds).

Table 5: Comparison of computational efficiency between PnP-DM and other baseline methods

| Inverse problem | Metric | DDRM | DPS | PnP-SGS | DPnP | PnP-DM (ours) |
|---|---|---|---|---|---|---|
| Super-resolution | Clock time (s) | 0.4 | 39 | 20 | 322 | 55 |
| | NFE | 20 | 1000 | 1030 | 18372 | 3032 |
| Coded diffraction patterns | Clock time (s) | – | 37 | 54 | 261 | 50 |
| | NFE | – | 1000 | 2572 | 14596 | 2482 |

# D   Implementation details of baseline methods

**PnP-ADMM**   We set the ADMM penalty parameter as 2 and ran for 500 iterations to ensure convergence. We used the pretrained DnCNN denoiser [81] available at the `deepinv` library[7].

**DPIR**   We followed the annealing schedule in [80] and ran for 40 iterations. We used the pretrained DRUNet denoiser [82] available at the `deepinv` library.

**DDRM**   We ran all the experiments with the default parameters: $\eta_B = 1.0$, $\eta = 0.85$, and 20 steps for the DDIM sampler [59]. For the Gaussian deblur problem, we used the SVD-based forward model implementation based on separable 1D convolution. We ran it with an additional batch dimension to collect multiple samples.

---

[7]https://github.com/deepinv/deepinv (BSD-3-Clause license)

**DPS** We followed the original paper to use a 1000-step DDPM sampler backbone. For the linear inverse problems, we used the step size given in [17], i.e. $\zeta' = 1$. For the nonlinear inverse problems, we optimized the step size $\zeta'$ by performing a grid search, which led to $\zeta' = 3$ for CDP and Fourier phase retrieval and $\zeta' = 0.001$ for black hole imaging. For the synthetic prior experiments, we also optimized the step size and used $\zeta' = 0.1$ for compressed sensing and $\zeta' = 1$ for Gaussian deblur. We ran it with an additional batch dimension to collect multiple samples.

**PnP-SGS** We performed a grid search for the coupling parameter $\rho$ and found that $\rho = 0.1$ worked the best for all problems. We followed the practice in [19] to have a burn-in period of 20 iterations during which the reverse diffusion is early-stopped. We ran the algorithm for 100 iterations in total and collect 20 samples in the 80 iterations after the burn-in period.

**DPnP** We implemented the `DDS-DDPM` sampler for the prior step. For fair comparison, we used the same annealing schedule for the coupling strength (denoted as $\eta_k$ in [78]) as PnP-DM. We ran it for the same number of iterations for each inverse problem with the same way of collecting samples as our method.

**HIO** We set $\beta = 0.7$ and applied both the non-negative constraint and the finite support constraint. To mitigate the instability of reconstruction depending on initialization, we first repeatedly ran the algorithm with 100 different random initializations and chose the reconstruction that has the best measurement fit. Then we ran another 10,000 iterations with the chosen reconstruction to ensure convergence and report the metrics on the final reconstruction.

# E   Additional related works

**Image reconstruction with plug-and-play priors** Plug-and-Play priors (PnP) [69] is an algorithmic framework that leverages off-the-shelf denoisers for solving imaging inverse problems. Recognizing the equivalence between the proximal operator and finding the *maximum a posteriori (MAP)* solution to a denoising problem, PnP substitutes the proximal update in many optimization algorithms, such as ADMM [13, 56] and HQS [82, 80], with generic denoising algorithms, particularly those based on deep learning [49, 82, 80]. The PnP framework enjoys both convergence guarantees [65, 56] and strong empirical performance [76, 1] due to its compatibility with state-of-the-art learning-based denoising priors. Recent works have also proposed learning-based PnP frameworks that have direction interpretations from an optimization perspective [20, 26]. See [36] for a comprehensive review on the theory and practice of PnP.

**Posterior sampling with MCMC and learning priors** Learning-based priors have also been considered in the Bayesian context [35], where one seeks to sample the posterior distribution defined under a learned prior. An important technique is denoising score matching (DSM) [70], which connects image denoising with learning the score function of an image distribution. Based on DSM, prior works have incorporated deep denoising priors into MCMC formulations, particularly focusing the Langevin Monte Carlo and its variants as they involve the score function of the target distribution [40, 34, 41, 66, 54]. Recently, methods based on SGS have also gained increasing popularity [53, 19, 6, 27, 78]. Unlike PnP methods based on optimization, these sampling methods possess the ability to generate diverse solutions and quantify the uncertainty of solution space.

**Solving inverse problems with diffusion models** The remarkable performance of diffusion models [33, 63] on modelling image distributions makes them desirable choices as images priors for solving inverse problems. One popular approach is to leverage a pretrained unconditional model and modify the reverse diffusion process during inference to enforce data consistency [73, 18, 17, 60, 39, 61, 62, 8, 44, 84, 58]. Despite of the promising performance of these methods, they usually involve approximations and empirically driven designs that are hard to justify theoretically and may lead to inconsistent sample distributions. Another line of work learns task-specific models, which achieves higher accuracy at the cost of re-training models for new problems [4, 57, 44]. Methods based on Particle Filtering and Sequential Monte Carlo are also considered to ensure asymptotic consistency [11, 23, 74]. Diffusion models have also been considered as a prior for variational inference [28, 47] and pluy-and-play image reconstruction [32, 48].

# F    Additional experimental results

## F.1    Synthetic prior experiment

In addition to the compressed sensing experiment presented in the main paper, we show another comparison on a Gaussian deblurring problem in Figure 10. Here, the linear forward model $A \in \mathbb{R}^{m \times n}$ is a 2D convolution matrix with a Gaussian blur kernel of size $7 \times 7$ and standard deviation 3.0. Similar to the compressed sensing experiment, both methods yield accurate reconstructions of the mean. However, in terms of the posterior standard deviation, DPS exhibits a notable difference from the ground truth, whereas our method achieves a significantly more accurate result.

## F.2    Linear inverse problems

We provide visual comparisons for the Gaussian deblur and super-resolution problems in Figure 11.

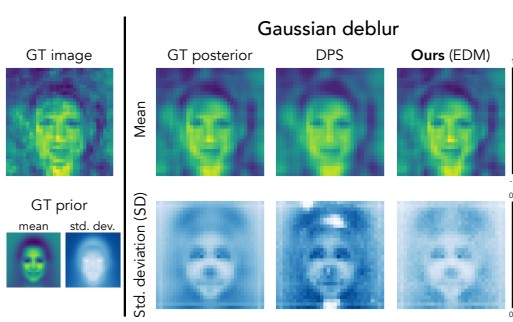

Figure 10: Comparison of our method and DPS [17] on estimating the posterior distribution of a Gaussian deblurring problem under a Gaussian prior. While the mean estimations of the two methods are of roughly the same quality, our approach provides a much more accurate estimation of the posterior per-pixel standard deviation than DPS.

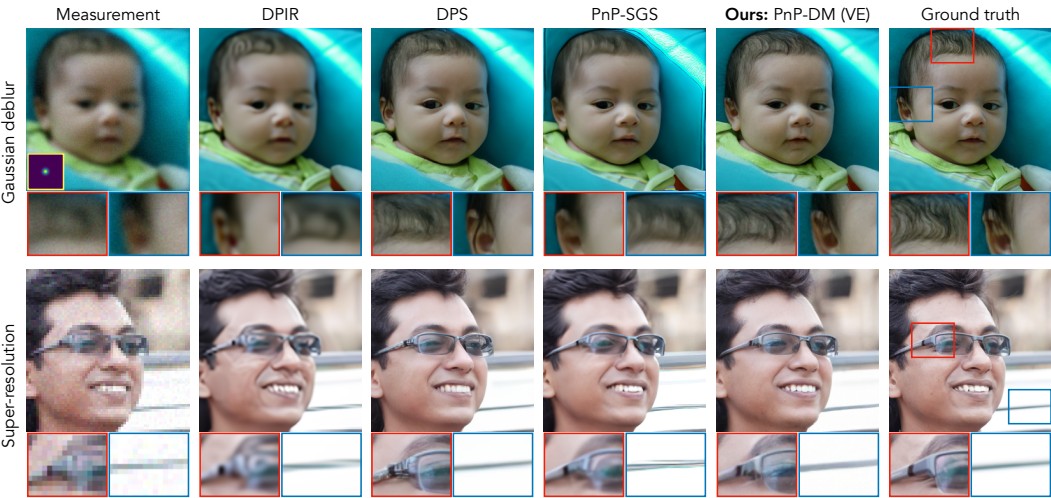

Figure 11: Visual comparison between our method and baselines on solving the Gaussian deblurring and super-resolution problems with i.i.d. Gaussian noise ($\sigma_y = 0.05$). We visualize one sample generated by each algorithm.

Additional visual examples are provided in Figure 12 (Gaussian deblurring), Figure 13 (motion deblurring), and Figure 14 (super-resolution).

## F.3    Nonlinear inverse problems

We provide visual comparisons for the CDP reconstruction problem in Figure 15, where we visualize one sample for each method. As shown by the red zoom-in boxes, PnP-DM can recover fine-grained features such as the hair threads that are missing in the reconstructions by the baselines. Additional reconstruction examples are given in Figure 16 for the CDP reconstruction problem.

We then show some additional reconstruction examples with comparison to DPS in Figure 17. For each method, we visualize the best reconstruction out of four runs for each test image according to the PSNR value. While DPS failed on around half of the test images, our proposed method provided high-fidelity reconstructions on almost all test images. This comparison highlights the better robustness of our method over DPS.

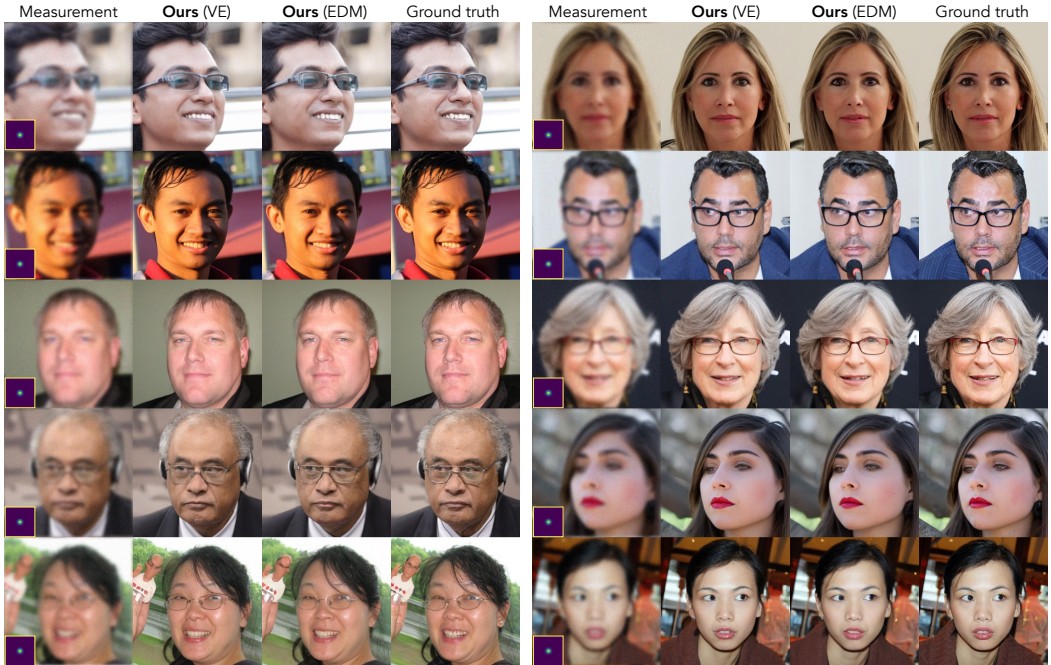

Figure 12: Additional visual examples for the Gaussian deblurring problem.

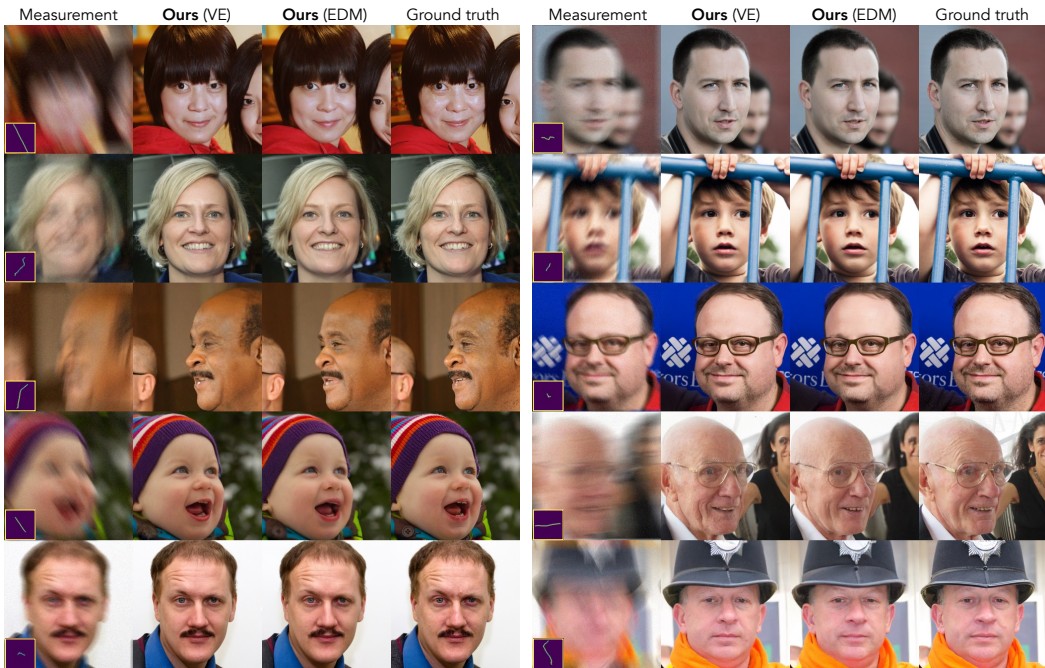

Figure 13: Additional visual examples for the motion deblurring problem.

## F.4 Black hole imaging

Finally, we present visual examples from the black hole imaging experiments. Samples generated by PnP-DM (EDM) and DPS using the simulated data are shown in Figure 18, with the data mismatch metric labeled at the top right corner of each sample. Consistent with the results in Figure 8, DPS can only capture one of the two posterior modes. DPS samples from Mode 2 and Mode 3 significantly deviate from the measurements and lack the expected black hole structure. In contrast, PnP-DM successfully samples both posterior modes and consistently produces samples that fit the measurements well. Additionally, Figure 19 presents more samples obtained by applying PnP-DM to the real M87 black hole data. The generated samples are not only diverse but also fit the measurements

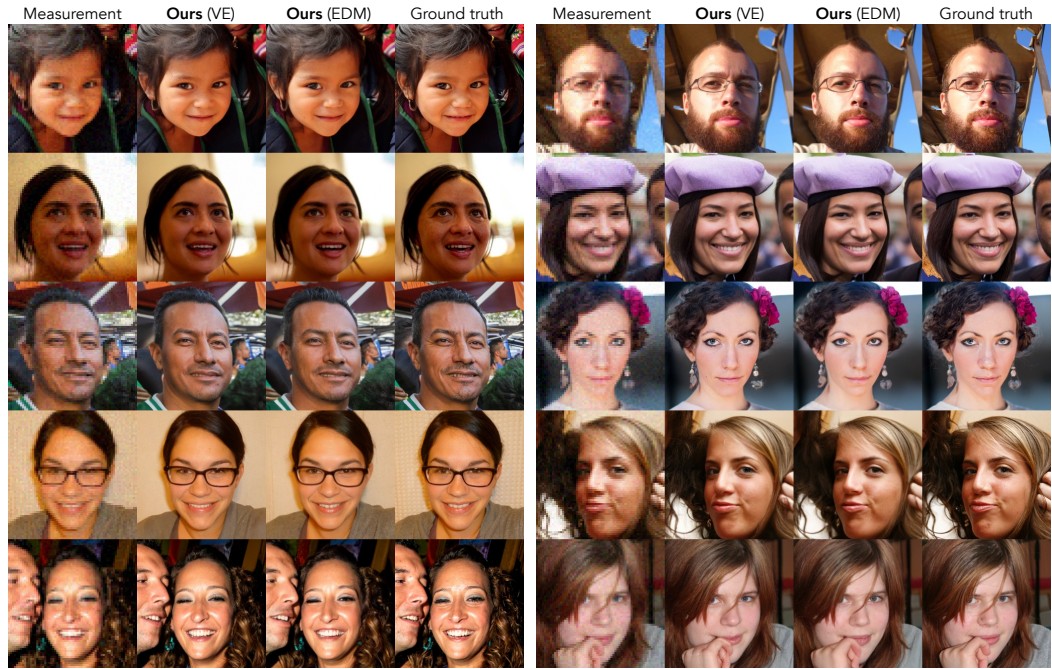

Figure 14: Additional visual examples for the 4× super-resolution problem.

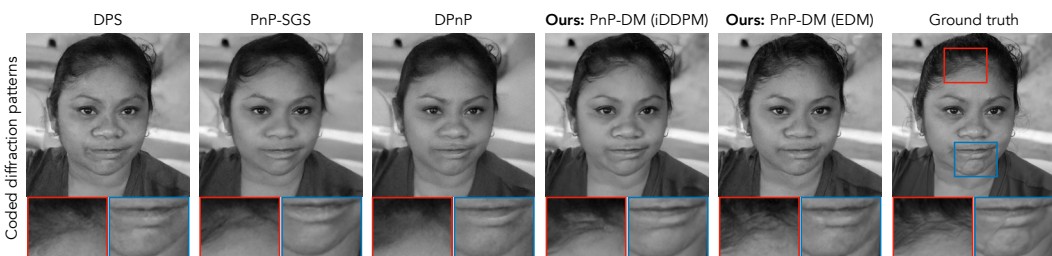

Figure 15: Visual comparison between our method and baselines on solving the coded diffraction pattern (CDP) reconstruction problems with i.i.d. Gaussian noise ($\sigma_{\boldsymbol{y}} = 0.05$). We visualize one sample generated by each algorithm.

well with data mismatch values around 2. These samples exhibit a ring diameter consistent with the official EHT reconstruction in Figure 1 and share a common bright spot location at the lower half of the ring.

### F.5 Further analysis

**Sensitivity analysis on the annealing schedule** $\{\rho_k\}$ In Figure 20, we present a sensitivity analysis on the annealing schedule $\{\rho_k\}$. In particular, we show the PSNR curves of $\boldsymbol{x}_k$ with different exponential decay rates $\alpha$ (left) and minimum coupling levels $\rho_{\min}$ (right) for one linear (super-resolution) and one nonlinear (coded diffraction patterns) problem. We have the following conclusions based on the results. First, different decay rates lead to different rates of convergence, which corroborates with our theoretical insights that $\rho$ plays the same role as the step size. The final level of PSNR is not sensitive to different decay rates, as all curves converge to the same level. Second, as $\rho_{\min}$ decreases, the final PSNR becomes higher. This is as expected because the stationary distribution of the $\boldsymbol{x}_k$, $\pi^X$ should converge to the true target posterior, $p(\boldsymbol{x}|\boldsymbol{y})$, as $\rho$ decreases.

**Convergence curves with intermediate visual examples** In Figure 21, we show some visual examples of intermediate $\boldsymbol{x}_k$ and $\boldsymbol{z}_k$ iterates (left) and convergence plots of PSNR, SSIM, and LPIPS for $\boldsymbol{x}_k$ (right) on the super-resolution problem. As $\rho_k$ decreases, $\boldsymbol{x}_k$ becomes closer to the ground truth and $\boldsymbol{z}_k$ gets less noisy. Both the visual quality and metric curves stabilize after the minimum

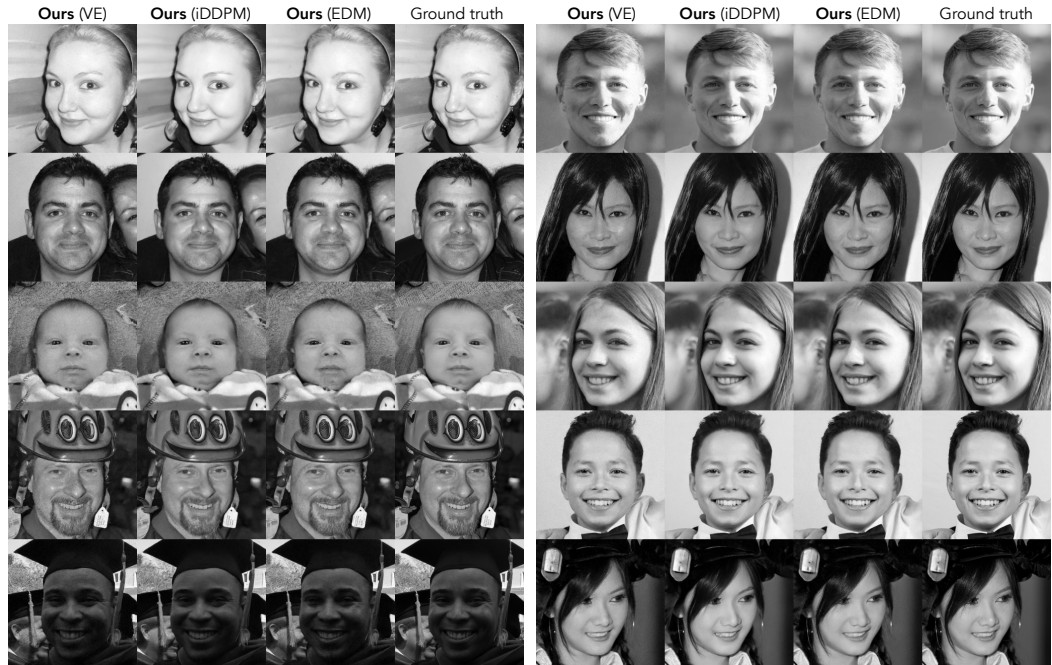

Figure 16: Additional visual examples for the Fourier phase retrieval problem.

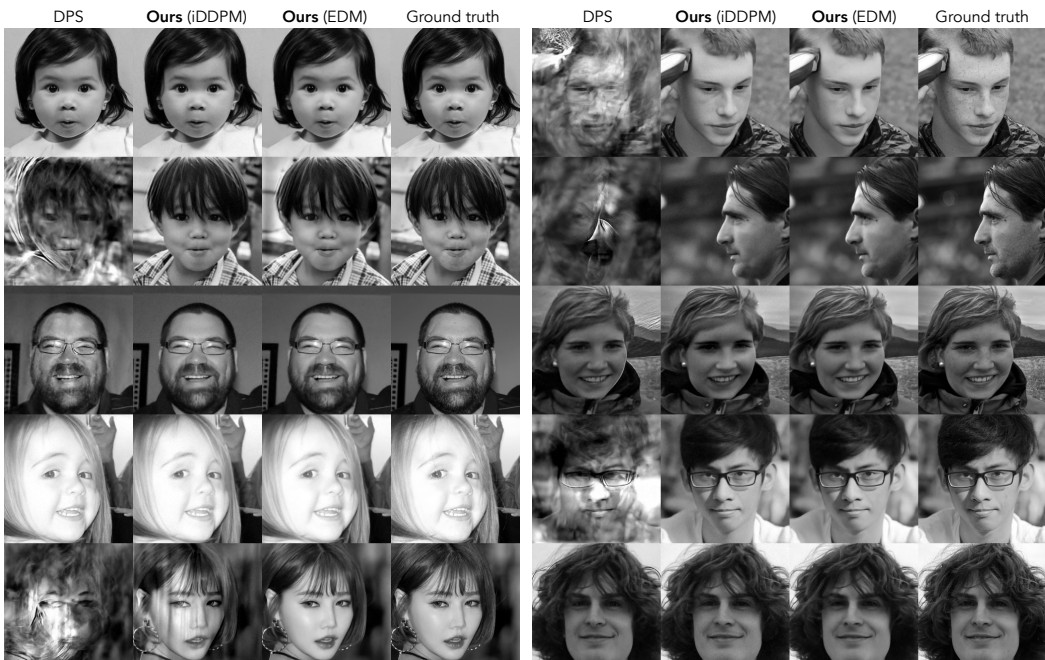

Figure 17: Additional visual examples for the Fourier phase retrieval problem.

coupling strength is achieved. Despite being run for 100 iterations in total, our method generates good images in around 40 iterations, which is around 30 seconds and 1600 NFEs.

## G   Licenses

We list the licenses of all the assets we used in this paper:

- Data
    - CelebA [45]: Unknown

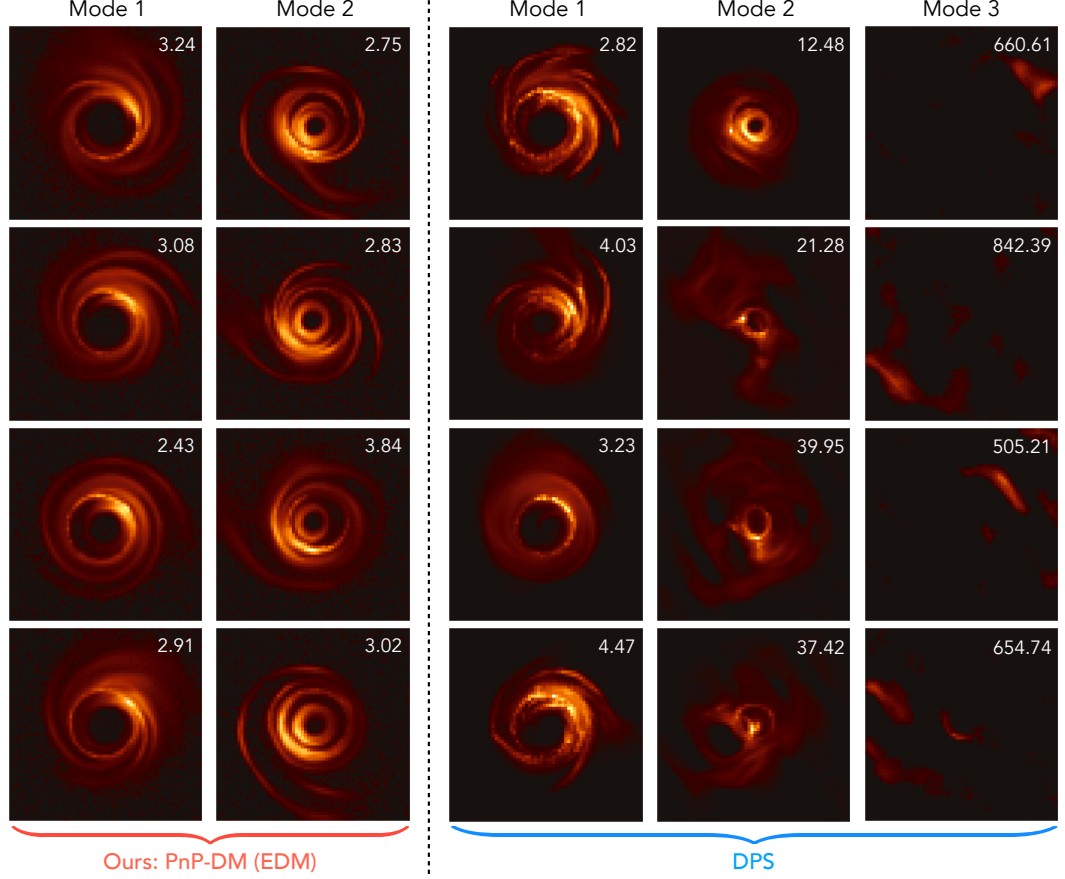

Figure 18: Additional visual examples given by PnP-DM and DPS using the simulated black hole data.

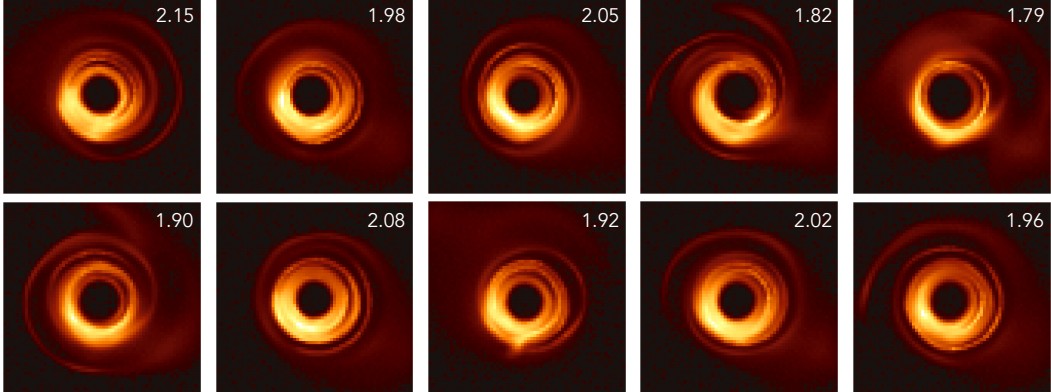

Figure 19: Additional visual examples given by PnP-DM using the real M87 black hole data.

- FFHQ [38]: Creative Commons BY-NC-SA 4.0 license
- Simulated and real black hole data: Unknown
- Code
  - The license for each code repository that we have used is listed in the footnote after the repository link.
- Pretrained model
  - FFHQ model by Choi et al. [16]: MIT license

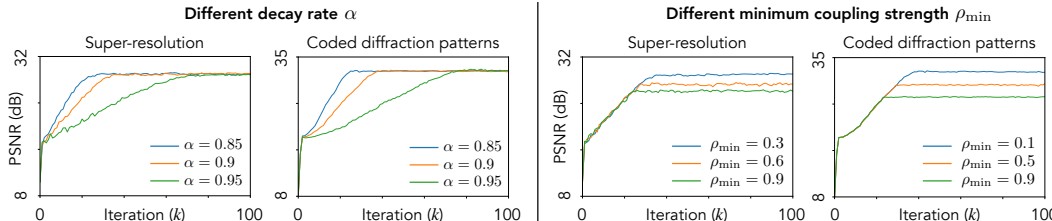

Figure 20: Sensitivity analysis on the annealing schedule $\rho_k$ with different decay rates $\alpha$ (left) and minimum coupling strength $\rho_{\min}$ (right) for a linear (super-resolution) and a nonlinear (coded diffraction patterns) inverse problem. Recall from Appendix C.3 that $\rho_k := \max(\alpha^k \rho_0, \rho_{\min})$, where we set $\rho_0 = 10$ for this experiment.

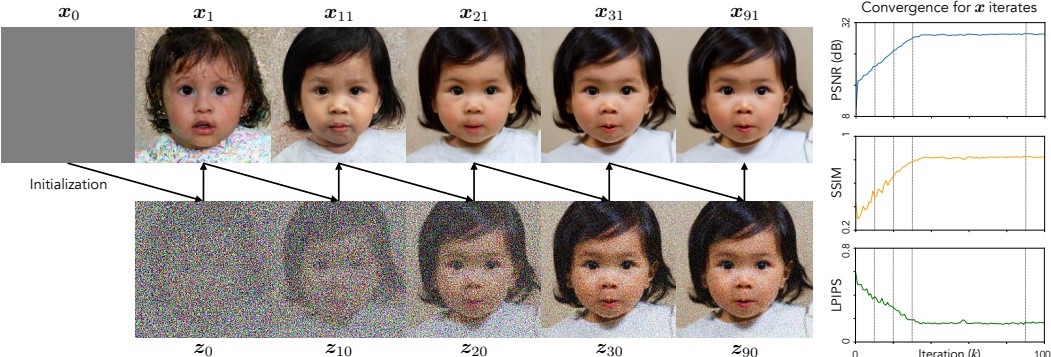

Figure 21: Visual examples of intermediate $\boldsymbol{x}_k$ and $\boldsymbol{z}_k$ iterates (left) and convergence plots of PSNR, SSIM, and LPIPS for $\boldsymbol{x}_k$ iterates (right) on the super-resolution problem. The vertical dashed lines show the iterations at which the $\boldsymbol{x}_k$ and $\boldsymbol{z}_k$ iterates are visualized.

