# OpenReview forum: "Principled Probabilistic Imaging using Diffusion Models as Plug-and-Play Priors"
_NeurIPS.cc/2024/Conference — NeurIPS 2024 poster_

### Official Review · Reviewer_43sJ · 2024-07-11

**Soundness:** 3
**Presentation:** 3
**Contribution:** 3
**Rating:** 6
**Confidence:** 3

**Summary:**

It is proposed to solve image inverse problems with diffusion priors by sliced Gibbs sampling (SGS). To sample from the product of prior and likelihood distributions, SGS assumes a pair of variables $x,z$ coupled by a Gaussian and alternates conditional sampling steps: on $x$ using the prior density multiplied with the conditional Gaussian and on $z$ using the likelihood multiplied with the conditional Gaussian. When the prior is a diffusion model, the two steps can be implemented by modifying the given pretrained score model and by Langevin dynamics (or exactly in the case of a linear problem), respectively. This procedure is evaluated on a few standard linear and nonlinear IPs and on a black home imaging problems, and the reconstructions found by the proposed method are typically visually and quantitatively better than the ones found by algorithms from prior work.

**Strengths:**

- The writing is mostly clear; I have no complaints on the paper organization or exposition.
  - In particular, the exposition of SGS in the general case followed by its application to DM priors is helpful.
- As far as I know, this is an original way of solving the diffusion posterior sampling problem.
- Strong results (if we are to accept that they support the claims) and application to a real imaging problem.
- Code is provided.

**Weaknesses:**

- On fairness and strength of experimental comparisons:
  - The results do not come with confidence intervals, which makes it hard to assess significance.
  - I may have missed/misunderstood this, but are all methods using the same pretrained prior? Appendix D seems to suggest otherwise, which would be a problem (it isn't fair to compare posterior sampling methods with different priors).
    - Similarly, different methods seem to be assuming different numbers of sampling steps and otherwise incompatible choices. How were hyperparameters, such as the $\rho_k$ annealing schedule, chosen?
  - Computation cost is not sufficiently discussed.
    - The method is run for at least a hundred Gibbs iterations in each experiment, and each one involves diffusion sampling initialized from some intermediate noise level. This makes the number of model evaluations quite large. It would be necessary to compare the number of function calls, as well as wall time, for all the methods.
- Convergence rate:
  - Theorem 3.1 tells us the rate of convergence to the stationary process for a fixed $\rho$, but not how close the stationary marginal is to the true posterior. They are not equal, in general, for positive $\rho$, and the rate of convergence shown has a $\frac1\rho$ factor, which means the convergence guarantees with small $\rho$ are weaker.
  - To understand the method, it would be important to show the dependence on the number of iterations, both by showing examples at different iterations in an illustrative case and by tracking the convergence of some metric at a function of the number of steps.
- Please also see questions below.

**Questions:**

- After equation (1), writing $y=A(x)+n$ where $n\in\mathbb{R}^n$ means that $y$ depends deterministically on $x$. In fact $n$ should be a random variable taking values in $\mathbb{R}^n$, which makes $y$ also a random variable.
- I am surprised not to see any reference to the paper whose title is a suffix of this one's, ["Diffusion models as plug-and-play priors", NeurIPS'22], also abbreviated "PnP" in some subsequent work. There a stochastic optimisation is used to find modes of the posterior.
  - Related to this, "PnP-DM" is not a very informative name for the proposed algorithm when so many of the baseline methods already have "PnP" in the name in some other configuration of letters.

**Limitations:**

Yes, but it would be good to compare computation costs, see above.

---

> ### Author Rebuttal · Authors · 2024-08-07
>
> Thank you for reviewing our paper. Below we provide point-by-point responses to your comments.
>
> > Weakness 1.1
>
> We provide the confidence intervals of all the results in Tables 1 and 2 in the attached PDF. Based on these results, we claim that our method achieves comparable or superior outcomes on most tasks, particularly for complex nonlinear inverse problems such as Fourier phase retrieval. Compared to the baseline methods, our standard deviation is slightly lower, indicating comparable and slightly better robustness of our approach.
>
> To better show the significance of the improvement of our method over baselines, we further conducted a sample-wise PSNR comparison between PnP-DM (EDM) and the most competitive baseline DPnP. The table below shows the sample-wise improvement rate PnP-DM over DPnP and $p$-value of the one-sided t-test for each inverse problem. Note that our method improves on *all samples in the test set* and the $p$-values indicate high statistical significance.
>
> **Response Table 2:** Sample-wise comparison between our method and DPnP [2].
>
> |                  | Gaussian deblur | Motion deblur | Super-resolution | Coded diffraction patterns | Fourier phase retrieval |
> |------------------|-----------------|---------------|------------------|----------------------------|-------------------------|
> | Improvement rate | 100%            | 100%          | 100%             | 100%                       | 100%                    |
> | t-test $p$-value | 8.18e-8         | 1.15e-6       | 9.40e-11         | 3.43e-70                   | 2.28e-2                 |
>
> > Weakness 1.2
>
> As mentioned in Page 7 line 244-246, we used the same model checkpoints for all diffusion model-based methods, which includes DDRM, DPS, PnP-SGS, DPnP, and our PnP-DM. This suggests that these sampling methods should share the same target posterior distribution.
> For our method, we fine-tuned the parameters in the annealing schedule $\rho_0$, $\alpha$, and $\rho_\min$ (notations from Appendix C.3) by performing a grid search over 20 FFHQ images outside of the test set we used for final reporting. Please also see the response to Question 2 by Reviewer u5tS.
>
> > Weakness 1.3
>
> Please refer to the response to all reviewers above for a comparison on computational efficiency. To clarify, as we mentioned in Appendix C.2 "Pseudocode" paragraph, the total number of time steps for the *entire* diffusion process is 100, but we only start running from some intermediate noise level. Therefore, the number of NFEs for each iteration is less than 100. In fact, according to the table, the average number of NFEs is only around 30 per iteration given the current annealing schedule.
>
> > Weakness 2.1
>
> Although we agree that there is a gap between the stationary process and the true posterior, we would like to argue that the convergence guarantees with small $\rho$ are not weaker and that the $\frac{1}{\rho}$ factor is in fact expected. For SGS-base methods, $\rho$ can be interpreted as the "step size." The step size has a similar presence in the bound of prior non-asymptotic analysis on the convergence of Langevin-based MCMC algorithm [8], [9]. For these algorithms, it is common to have an $O(1/T)$ convergence rate where $T$ is the total diffusion time and scales with the step size, so there will be a $\frac{1}{\text{step size}}$ factor on the right hand side. This also justifies the annealing strategy, which starts with large $\rho$ (hence large step size) at the beginning when the distribution is far away from the target and gradually decreases.
>
> > Weakness 2.2
>
> Thank you for your suggestion. In Figure 2 of the attached PDF file, we show some visual examples of intermediate $\boldsymbol{x}$ and $\boldsymbol{z}$ iterates (left) and convergence plots of PSNR, SSIM, and LPIPS for $\boldsymbol{x}$ iterates (right) on the super-resolution problem. As $\rho_k$ decreases, the $\boldsymbol{x}$ iterate becomes closer to the ground truth and the $\boldsymbol{z}$ iterate gets less noisy. Both the visual quality and metric curves stabilize after the minimum coupling strength $\rho_{\min}$ is achieved. Despite being run for 100 iterations in total, our method generates good images in around 40 iterations, which is around 30 seconds and 1600 NFEs. We can include these results in our supplemental material of the revised manuscript.
>
> > Question 1
>
> We will change the writing on this part in the final version of the paper.
>
> > Question 2
>
> Thank you for bringing this paper to our attention. We notice that [one reviewer's comment](https://openreview.net/forum?id=yhlMZ3iR7Pu&noteId=IRfkHFMGG6#:~:text=for%20downstream%20tasks.-,The%20naming%20of%20the%20framework%20is%20confusing.%20Plug%2Dand%2Dplay%20priors%20(PnP)%20is%20a%20well%2Dknown%20framework%20in%20the%20literature%20of%20image%20restoration%20and%20imaging%20inverse%20problems.%20The%20title%20confuses%20me%20in%20the%20first%20place.,-The%20naming%20is) of this work also touches on the naming of the method. According to the authors' response, this happens to be a clash of terminology. However, we will include this paper in the related work. As for the "PnP-DM" name, we understand that "PnP" is so commonly used that it may cause some ambiguity. We used it mainly to highlight its similarity to deterministic PnP methods that incorporate an image prior via denoising.

---

> > ### Comment · Reviewer_43sJ · 2024-08-12
> >
> > I'm happy to recommend acceptance given the clarifications. Thank you.

---

### Official Review · Reviewer_TmFe · 2024-07-12

**Soundness:** 4
**Presentation:** 4
**Contribution:** 4
**Rating:** 7
**Confidence:** 5

**Summary:**

This paper proposes a novel posterior sampling algorithm for solving inverse problems with Diffusion models. The proposed algorithm is based on the Split Gibbs sampling scheme and consists in, first introducing an extended distribution $\pi_\rho(x, z)$ that admits the original posterior $\pi$ as $x$ marginal after letting $\rho$ tend to $0$. Then, for a fixed $\rho$ this joint distribution can be sampled via Gibbs sampling; the form of the joint distribution $\pi_\rho(x, z)$ allows sampling sampling both $X$ and $Z$ easily. For $Z$, this can be done exactly in the case of linear inverse problems, or with Langevin Monte Carlo otherwise. For $X$, the particular structure of the joint distribution is exploited and as it turns out, it can be sampled by simply running the learned backward Diffusion. Notably, this is the only step that requires using the learned denoiser. The theoretical properties of the ideal sampler are inverstigated and extensive experiments are considered.

**Strengths:**

I highly enjoyed going through this paper and I found it to be interesting. The strengths are:
- The method is original and clever; the joint distribution introduced, which stems from the SGS approach, allows almost exact
sampling of both the conditional distributions. Furthermore, the considered framework allows having actual theoretical guarantees for the sampler (although in an ideal case). This is still quite informative and contrasts with existing approaches (besides the SMC-based ones which also allow more straightforward theoretical guarantees). The main perk of this method wrt to the latter is that it has theoretical guarantees while not requiring absurd memory requirements due to the storage of a large number of particles.

- The writing is extremely clear. The paper is written in a way that It is straightforward to understand while still being quite detailed. This is a significant perk (Still, i think that section 3.4 could be improved as I had to go through the supplementary material to actually understand what the proof technique was).

- Finally, the experiments are very good; the authors first consider a toy example in which the posterior has a closed form. This serves as a nice introduction to the experimental section and showcases the perks of the present method in comparison with DPS. Next, the imaging experiments are rigorously executed, with a precise look at the uncertainty quantification. Finally, an original black hole experiment is considered and I found it to be very interesting.

**Weaknesses:**

- **Computational cost**:  I believe that the proposed method is much slower than existing alternatives. This is not that much of an issue for me (but of course this depends on the reader and practitioner). In my opinion the compute time is not properly addressed; although the authors discuss it in the bottom of page 21, I would like to see a proper runtime comparison with existing methods. I would like to emphasize that  this is only a minor weakness.

- **Originality**: I believe that the proposed method is related to SDEdit [1]; it can be thought of as a variant of SDEdit. Indeed, to see why this is the case, when the present algorithm is applied to inpainting, the first iteration, the likelihood step, will result in a noisy image with the observation already present In it. This image is then denoised using the backward process during the prior step. This is in fact very similar to sdedit which starts with some initial image, say the pseudo-inverse of the observation, noises it to some given noise level and denoises it using the backward Diffusion process. See Algorithm 3, [1]. I believe that the main difference with the algorithm presented in your paper is the use of the decreasing noise schedule, which makes sense theoretically. For a fair treatment I suggest you mention this in your paper.

[1] Meng, C., He, Y., Song, Y., Song, J., Wu, J., Zhu, J.Y. and Ermon, S., 2021. Sdedit: Guided image synthesis and editing with stochastic differential equations.

**Questions:**

see above

**Limitations:**

see above

---

> ### Author Rebuttal · Authors · 2024-08-07
>
> Thank you for your evaluation of our work. Below we provide point-by-point responses to your comments.
>
> > **Computational cost**: I believe that the proposed method is much slower than existing alternatives. This is not that much of an issue for me (but of course this depends on the reader and practitioner). In my opinion the compute time is not properly addressed; although the authors discuss it in the bottom of page 21, I would like to see a proper runtime comparison with existing methods. I would like to emphasize that this is only a minor weakness.
>
> We provide a comparison on computational efficiency for all methods in Response Table 1 above. Our method PnP-DM achieves superior performance while remaining comparable in runtime with DPS, with a runtime no more than 1.5 times that of DPS. In fact, for the linear inverse problem experiments, PnP-DM was faster than DPS because we were able to take multiple samples along one Markov chain generated by PnP-DM but DPS requires running multiple diffusion processes.
>
> > **Originality**: I believe that the proposed method is related to SDEdit [7]; it can be thought of as a variant of SDEdit. Indeed, to see why this is the case, when the present algorithm is applied to inpainting, the first iteration, the likelihood step, will result in a noisy image with the observation already present In it. This image is then denoised using the backward process during the prior step. This is in fact very similar to sdedit which starts with some initial image, say the pseudo-inverse of the observation, noises it to some given noise level and denoises it using the backward Diffusion process. See Algorithm 3, [7]. I believe that the main difference with the algorithm presented in your paper is the use of the decreasing noise schedule, which makes sense theoretically. For a fair treatment I suggest you mention this in your paper.
>
> Thank you for bringing this work to our attention. This work indeed shares some similarity to our own in that it employs a process of noising followed by denoising. Nevertheless, our method is more theoretically grounded and meant for rigorously estimating the posterior of an inverse problem, whereas SDEdit is more empircally driven and designed for solving image synthesis/editing problems. We will cite and discuss [7] in the related work section in the final version.

---

> > ### Comment · Reviewer_TmFe · 2024-08-12
> >
> > Thank you for your response.
> >
> > Reviewer u5tS mentions that this work bears significant similarities with the work of Coeurdoux et al. and I agree that this is the case with the main difference being that they use a constant $\rho$, which is also more or less the main difference between your work and that of [1] that I am mentioning in my initial review. While reading the paper I did not notice these similarities with the work of Coeurdoux et al., this means that paper in its current form does not adequately address the related work and might mislead the reader into thinking that the paper is much more novel than what it actually is.
> >
> > Nonetheless, the empirical results are promising and show that the decreasing schedule is indeed a good idea. I maintain my score but i strongly recommend the authors to modify their paper and clearly state the similarities with previous works.

---

> > > ### Author Response · Authors · 2024-08-13
> > >
> > > Thank you for your response. We will provide additional clarifications on the similarities and differences between our work and prior works in the final version. Please also see [responses W1.1--W1.3](https://openreview.net/forum?id=Xq9HQf7VNV&noteId=O9TzVFCQmB) to Reviewer u5tS for more information on other differences.

---

### Official Review · Reviewer_G2Ty · 2024-07-12

**Soundness:** 4
**Presentation:** 3
**Contribution:** 4
**Rating:** 8
**Confidence:** 3

**Summary:**

The paper introduces a method to address Bayesian inverse problems in computational imaging. It leverages the generative capabilities of diffusion models (DMs) to sample the posterior distribution over all possible solutions from noisy and sparse measurements. The method combines a Markov chain Monte Carlo (MCMC) algorithm with a general DM formulation to perform rigorous posterior sampling, effectively integrating state-of-the-art DMs as expressive image priors. The method deviates from current methods that rely on approximating the intractable posterior via separating the forward operator from an unconditional prior over the intermediate noisy image. The approach is validated on six inverse problems, demonstrating superior accuracy and posterior estimation compared to existing DM-based methods.

**Strengths:**

- The paper presents a novel combination of MCMC and diffusion models to solve inverse problems. This use of DMs as priors in a Bayesian framework can be integrated into SOTA diffusion models as plug-n-play expressive image priors for Bayesian inference.

- The presented method is rigorous. The use of the Split Gibbs Sampler and the EDM formulation for the prior step is well-executed. The theoretical insights provided, including the stationarity guarantee in terms of average Fisher information, add depth to the method.

- The paper is well-structured and supported by diagrams and pseudocode to enhance understanding. The distinction between existing methods and the proposed approach is clearly articulated.

- Experiments show the efficacy of the proposed method compared to existing methods on a diverse set of real-world problems, highlighting its potential broader impact on computational imaging applications.

**Weaknesses:**

- While the method demonstrates superior performance, the computational cost and efficiency are not thoroughly discussed. A comparison of computational resources required compared to other methods would be beneficial.

The impact of various parameters, such as the annealing schedule for the coupling parameter ρ, on the method's performance is not deeply investigated. A sensitivity analysis could provide insights into the method's robustness.

**Questions:**

1. Can the authors compare the computational resources and time required for PnP-DM versus existing DM-based methods?

2. How sensitive is the performance of PnP-DM to the choice of the annealing schedule for the coupling parameter ρ? Is there an optimal range or strategy for selecting these parameters?

**Limitations:**

Yes.

---

> ### Author Rebuttal · Authors · 2024-08-07
>
> Thank you for your feedback on our work. Below we provide point-by-point responses to your comments.
>
> > While the method demonstrates superior performance, the computational cost and efficiency are not thoroughly discussed. A comparison of computational resources required compared to other methods would be beneficial. Can the authors compare the computational resources and time required for PnP-DM versus existing DM-based methods?
>
> We provide a comparison on computational efficiency for all methods in Response Table 1 above. Our method PnP-DM achieves superior performance while remaining comparable in runtime with DPS, with a runtime no more than 1.5 times that of DPS. In fact, for the linear inverse problem experiments, PnP-DM was faster than DPS because we were able to take multiple samples along one Markov chain generated by PnP-DM but DPS requires running multiple diffusion processes.
>
> > The impact of various parameters, such as the annealing schedule for the coupling parameter ρ, on the method's performance is not deeply investigated. A sensitivity analysis could provide insights into the method's robustness. How sensitive is the performance of PnP-DM to the choice of the annealing schedule for the coupling parameter ρ? Is there an optimal range or strategy for selecting these parameters?
>
> Following your suggestion, we have included a sensitivity analysis on the annealing schedule. In Figure 1 of the attached PDF file, we show the PSNR curves of different exponential decay rates $\alpha$ (left) and minimum coupling levels $\rho_{\min}$ (rate) for one linear (super-resolution) and one nonlinear (coded diffraction patterns) problem.
>
> We have the following conclusions based on the results. First, different decay rates lead to different rates of convergence, which corroborates with our theoretical insights that $\rho$ plays the same role as the step size. The final level of PSNR is not sensitive to different decay rates, as all curves converge to the same level. Second, as $\rho_{\min}$ decreases, the final PSNR becomes higher as the stationary distribution converges to the true target posterior.
>
> Our strategy for choosing the annealing parameters is as follows. The starting coupling strength $\rho_0$ should be large to overcome the ill-posedness of the problem (usually around 5 to 10 is sufficient). The minimum coupling strength $\rho_{\min}$ should be small to ensure that the stationary distribution is close to the target posterior; empirically around 0.1 to 0.3 works the best. The range of decay rate $\alpha$ is generally flexible; a value around 0.9 usually leads to good results.

---

> > ### Comment · Reviewer_G2Ty · 2024-08-12
> >
> > Thanks for the clarification and additional experiments. This makes sense to me.

---

### Official Review · Reviewer_G64m · 2024-07-14

**Soundness:** 4
**Presentation:** 4
**Contribution:** 4
**Rating:** 8
**Confidence:** 4

**Summary:**

This paper proposes a Markov Chain Monte Carlo algorithm for posterior sampling in both linear and non-linear inverse problems. The core of the proposed method is based on a Split Gibbs Sampler that alternates between two steps: one involving the likelihood and the other the prior. Additionally, the paper connects the Bayesian denoising problem with unconditional generation using Diffusion Models. The proposed method is validated on a range of linear and non-linear inverse problems, with an additional real-world application in black hole imaging.

**Strengths:**

- The proposed method outperforms DPS, with satisfactory evidence provided by the authors.
- Unlike some previous works, the proposed method is effective for both linear and non-linear inverse problems.
- The paper provides sufficient theoretical analysis to support the proposed method.

**Weaknesses:**

There are no apparent weaknesses. However, I am curious about the reconstruction speed comparison (seconds/image) between DPS and the proposed method. It would be practically very attractive for the community to use this algorithm if a rigorous comparison of reconstruction speed is made.

**Questions:**

No questions.

**Limitations:**

No limitations other than those mentioned in the paper.

---

> ### Author Rebuttal · Authors · 2024-08-07
>
> Thank you for your evaluation of our paper. We respond to your comment below.
>
> > There are no apparent weaknesses. However, I am curious about the reconstruction speed comparison (seconds/image) between DPS and the proposed method. It would be practically very attractive for the community to use this algorithm if a rigorous comparison of reconstruction speed is made.
>
> We provide a comparison on computational efficiency for all methods in Response Table 1 above. Our method PnP-DM achieves superior performance while remaining comparable in runtime with DPS, with a runtime no more than 1.5 times that of DPS. In fact, for the linear inverse problem experiments, PnP-DM was faster than DPS because we were able to take multiple samples along one Markov chain generated by PnP-DM but DPS requires running multiple diffusion processes.

---

> ### Comment · Reviewer_G64m · 2024-08-12
>
> Thank you for sharing the sampling speed results. The shared results make sense to me. I strongly feel that this paper provides novel enough contributions to be considered for acceptance. I have raised my confidence score to 4. Hope this helps AC decide about the paper.

---

### Official Review · Reviewer_u5tS · 2024-07-15

**Soundness:** 3
**Presentation:** 3
**Contribution:** 2
**Rating:** 4
**Confidence:** 4

**Summary:**

In this paper, the authors treat the problem of sampling from posterior of image inverse problems. Their formulation is based on the Split Gibbs Sampler (SGS) algorithm, which alternates between sampling from Moreau regularized versions of the prior and of the likelihood. The main contribution of the paper is to use the EDM formulation of diffusion models for sampling the prior term. The EDM formulation was
proposed to unify various diffusion models in a common general formulation.

**Strengths:**

Firstly the paper is well-written, provides clear and concise explanations of the methods and contributions. The authors have also included some theoretical insights that guarantee the convergence of each sampling step of the algorithm to the right stationary distributions when the number of iterations tends to infinity.

The strength of the paper is mainly its experimental section, which covers a range of both linear and non-linear inverse problems. Additionally, I find interesting the inclusion of a toy experiment with a Gaussian prior, which allows comparing to ground-truth posterior distributions.

**Weaknesses:**

1. **Significance of Contribution**: My primary concern is the overall significance of the contribution. When compared to the works of Coeurdoux et al. (2023) and Xu & Chi (2024), the advancements appear to be relatively minor. Specifically, the contribution seems to involve using a more general formulation of diffusion models (EDM) instead of the more specific DDPM or DDIM. Are there additional conceptual differences? Furthermore, given this similarity, what accounts for the significant performance disparity with these methods?

2. **Algorithm Efficiency**: The algorithms seem to require considerable computational time, necessitating the diffusion process to be run at each iteration. While this is mathematically sensible, it appears excessive. Given the annealing process chosen, how many calls to the denoiser are typically made in practice? Additionally, how does this number compare to methods that perform a single diffusion using an approximation of p(y|x_k), such as DPS? Including a comparative table in the paper would be valuable.

3. **Theoretical Results**: The theoretical results presented, while interesting, are somewhat limited as they assume a fixed rho parameter, which is not the case in practical applications. Additionally, the algorithm is presumed to run with the true score, which is also not realistic. In addition, although it may be beyond the scope of the current paper, providing theoretical insights into the convergence of the proposed Gibbs Sampler (with approximate score) to the true posterior would be beneficial.

**Questions:**

- In Figure 4, the diffusion process is represented in a closed-form. Did you train a denoiser, or did you use the closed-form score instead?

- How did you tune the hyperparameters for your method and for each method in the comparison? Did you use their default hyperparameters? For example, DPIR provides hyperparameters fine-tuned for various inverse problems. However, the SR x4 task is not one of the inverse problems for which DPIR's authors fine-tuned their hyperparameters, likely making the default hyperparameters sub-optimal for this task.

- Did you employ any regularization parameters to balance the data-fidelity and regularization terms?

- For the backward diffusion process, did you use the same number of iterations for all values of  $\rho_k$ ? Specifically, is the step size of the diffusion process consistent, or is it adjusted according to the value of $\rho_k$ ?

**Limitations:**

No limitations.

---

> ### Author Rebuttal · Authors · 2024-08-07
>
> Thank you for your efforts on reviewing our paper. Below we provide point-by-point responses to your comments:
>
> > Weakness 1
>
> We believe that our work has several significant contributions over the existing works [1] and [2]. Moreover, we highlight that according to [NeurIPS 2024 policy](https://neurips.cc/Conferences/2024/CallForPapers), our work should be considered as a concurrent work with [2] and "not be rejected on the basis of the comparison to contemporaneous work" as [2] was first published on arXiv on March 25th, 2024, which is within two months of the submission deadline of NeurIPS 2024.
>
>  - Compared to [1], the significance of our work lies in three aspects:
>      - Nonlinear inverse problems are beyond the scope of [1], while we have investigated three different nonlinear inverse problems.
>      - We have adopted a more rigorous formulation than [1]. As also pointed out by [2], the denoiser design based on diffusion models in [1] is heuristic.
>      - Unlike PnP-SGS [1], we consider an annealing strategy, which is important for ill-posed inverse problems.
>
> - Compared to [2], our work have the following three contributions:
>      - Our work proposes a more general formulation based on EDM. To the best of our knowledge, using EDM as a rigorous Bayesian denoiser for solving inverse problems has not been explored in prior literature.
>      - Our proposed method has a significant improvement in performance over DPnP [2]. Response Table 2 in the response to reviewer 43sJ below further shows the statistical significance of the improvement. One reason for the performance gain is that we inherited the optimized design choices from EDM [3], which provides better image quality with less diffusion steps. Another reason is that the flexibility of the framework provides a larger design space with more flexible parameter choices.
>
> > Weakness 2
>
> We provide a comparison on computational efficiency for all methods in Response Table 1 above. Our method PnP-DM achieves superior performance while remaining comparable in runtime with DPS, with a runtime no more than 1.5 times that of DPS. In fact, for the linear inverse problem experiments, PnP-DM was faster than DPS because we were able to take multiple samples along one Markov chain generated by PnP-DM but DPS requires running multiple diffusion processes.
>
> > Weakness 3
>
> - **Fixed $\rho$**. Our current analysis considers a fixed $\rho$ across different iterations, but we find that a variable schedule for $\rho$ is more practical. Theoretically, our analysis could be extended to the setting with a variable $\rho$ by viewing the iterations with larger $\rho$ values as warm-up iterations that produce favorable initial conditions for the algorithm with smaller $\rho$ values.
> - **Approximate score bound**. We can extend our convergence bound to the case where the learned score has error, which we can include in our revised manuscript. More precisely, assume we implement the EDM reverse diffusion step (as given by equation (10) in the manuscript) as: $$\mathrm{d} \boldsymbol{x}\_t=\left[u(t) \boldsymbol{x}\_t- v(t)^2 s\_t\left(\boldsymbol{x}\_t\right)\right] \mathrm{d} t + v(t)\mathrm{d}\bar{\boldsymbol{w}}\_t,$$
>
>     where $s\_t$ is the approximate score. Then, by differentiating the KL divergence along the two dynamics or using the Girsanov theorem (in the spirit of the proof in [4]), we can prove the following: for $\tau \in [k(1+t^\ast)+1, (k+1)(1+t^\ast)]$, which corresponds to the prior step in the Split Gibbs Sampler, we have:
>     $$\frac{1}{4}\int\_{k(1+t^\ast)+1}^{(k+1)(1+t^\ast)} \lambda(\tau)\text{FI}(\pi\_\tau\Vert \nu\_\tau)\mathrm{d}\tau \leq \epsilon\_{\text{score}}(k) + \text{KL}(\pi\_{(k+1)(1+t^\ast)}\Vert \nu\_{(k+1)(1+t^\ast)}) - \text{KL}(\pi\_{k(1+t^\ast)+1}\Vert \nu\_{k(1+t^\ast)+1})$$
>
>     where $\epsilon\_{\text{score}}(k) := \int\_{0}^{t^\ast} v(t)^2\mathbb{E}||s\_t(\boldsymbol{x}\_t^{(k)}) - \nabla \log p\_t(\boldsymbol{x}\_t^{(k)})||^2 \mathrm{d} t$ and $\boldsymbol{x}\_t^{(k)}, 0\leq t \leq t^\ast$ is the exact EDM reverse process, which satisfies equation (10) in the manuscript, starting at $\boldsymbol{x}\_0^{(k)} \sim \nu\_{k}^Z$ and ending up at $\boldsymbol{x}\_{t^\ast}^{(k)} \sim \nu\_{k+1}^X$. Therefore, up to score approximation errors, a convergence bound analogous to Theorem 3.1 will hold.
> - **Convergence to the true posterior**. Indeed, the asymptotic convergence to the true posterior as $\rho\to 0$ is difficult to obtain (see Section 5 of [5]). We agree that this is out of the scope of the paper, so we leave it to future work.
>
> > Question 1
>
> We trained a denoiser for the Gaussian image prior. More details are presented in Appendix C.2 "Model checkpoint" paragraph.
>
> > Question 2
>
> Please refer to Appendix D for how we chose the hyperparameters for all the methods. For our method, we fine-tuned the parameters in the annealing schedule $\rho_0$, $\alpha$, and $\rho_\min$ (notations from Appendix C.3) by performing a grid search over 20 FFHQ images outside of the test set we used for final reporting (referred to as the validation set hereafter). For DDRM and DPS on linear inverse problems, we used the default parameters in their official repositories. For all other cases, we performed a grid search of the main parameter(s) of each method on the validation set. Specifically, for DPIR, we used the same annealing function as that in the official repository of [6] but fine-tuned the starting/ending noise level and number of iterations.
>
> > Question 3
>
> We did not employ any explicit regularization parameters. Within the Bayesian framework, the data-fidelity (likelihood) and regularization (prior) should be automatically balanced based on the noise distribution.
>
> > Question 4
>
> The discretization time steps are the same for all values $\rho_k$. The number of iterations that are actually run depends on the specific value of $\rho_k$. See Appendix C.2 “Pseudocode” paragraph and Algorithm 3 for more details.

---

> > ### Comment · Reviewer_u5tS · 2024-08-08
> > **Response to rebuttal**
> >
> > I thank the authors for their detailed answers. Here are additional remarks / questions.
> >
> > **W1:**
> >
> > 1. Although not demonstrated in the experiments, I believe the approach in [1] could be equally applied to nonlinear inverse problems.
> >
> > 2. Could you clarify why you consider the denoiser design based on diffusion models in [1] to be heuristic?
> >
> > 3. The annealing strategy, despite not being adopted in [1], is commonly used in Plug-and-Play literature. However, incorporating this strategy introduces a discrepancy between your theoretical framework and experimental results, which is concerning.
> >
> > 4. I acknowledge that using EDM as the diffusion model offers a more flexible and optimized framework.
> >
> > **W2:**
> >
> > I do not understand why the clock time does not scale linearly with NFE. It appears that the operations outside the denoiser evaluation in both DPS and your method are relatively insignificant in time.
> >
> > **W3:**
> >
> > 1. I strongly disagree with your statement, as you did not run the algorithms with fixed $\rho$ even for smaller values.
> >
> > 2. Can you explicitly state the final convergence bound analogous to Theorem 3.1, including the time integral from 0 to T, that would result from this additional theoretical consideration?

---

> ### Author Response · Authors · 2024-08-09
>
> Thank you for your response and comments. Below we answer them point by point.
>
> > **W1.1**
>
> Indeed, we have tried to apply PnP-SGS [1] to nonlinear inverse problems by combining our likelihood step with its prior step design, which was how we obtained the PnP-SGS results for Table 2. While it could still handle the coded diffraction pattern problem, it failed on the more challenging Fourier phase retrieval problem. We also tried to include the annealing strategy for PnP-SGS but found that the method diverged with large $\rho$, probably due to its heuristic design of the prior step, and thus did not benefit from annealing. Overall, our method significantly outperforms PnP-SGS by at least 1dB in PSNR for all linear problems and coded diffraction patterns, and 15dB for Fourier phase retrieval. Our experimental results indicate that PnP-SGS struggles with challenging nonlinear inverse problems, such as Fourier phase retrieval.
>
> > **W1.2**
>
> There are mainly two aspects. Here we explain using the notations in the original DDPM paper. First, to rigorously implement the prior step of SGS, one need to assume the input as an observation of the *unscaled* image $\boldsymbol{x}_0$ with additive white Gaussian noise. However, in DDPM, the mean of state $\boldsymbol{x}\_t$ is not $\boldsymbol{x}\_0$, but $\sqrt{\bar{\alpha}\_t}\boldsymbol{x}\_0$, which is a *down-scaled* version of $\boldsymbol{x}\_0$. So, it is inaccurate to directly use DDPM as a denoiser. This mismatch is particularly significant when starting from a large $t^\ast$ as $\sqrt{\bar{\alpha}\_{t^\ast}}$ is close to 0. Second, according to the SGS formulation, the noise estimation module is unnecessary, and one should always denoise $\boldsymbol{z}^{(k)}$ (the $\boldsymbol{z}$ iterate of SGS at iteration $k$) assuming a noise level $\rho$. However, PnP-SGS does not take into account the hyperparameter $\rho$ in the denoising problem. It is unclear how the generated $\boldsymbol{x}^{(k+1)}$ relates to the target conditional distribution $\pi^{X|Z=\boldsymbol{z}^{(k)}}(\boldsymbol{x})$ for each prior step of SGS. Therefore, the posterior distribution sampled from the denoiser in [1] is not the desired posterior distribution even with a perfect score function.
>
> > **W1.3**
>
> We never claimed that the idea of annealing is our contribution. Nevertheless, we believe that it is our contribution to propose a general framework that accommodates annealing in an easy yet rigorous way for SGS and leads to significant performance improvement. Our focus is more on the algorithm design and experimental validation. To the best of our knowledge, no existing theory on SGS provides a non-asymptotic convergence bound with annealing, so this is a a challenging open question. Although our current theory assumes fixed $\rho$, it still provides some theoretical insights on the algorithm behavior, such as the interpretation of $\rho$ as step size, and potentially opens up new theoretical directions for SGS based on the Fisher information. We hope to extend the analysis to the annealing case in the future.
>
> > **W2**
>
> We respectfully disagree with your last statement. DPS requires backpropagating through the entire denoiser network for each diffusion step after the forward pass of the denoiser, introducing a significant computational overhead. On the other hand, our method does not require doing so. This is the main reason why the clock time does not scale linearly with NFE for these two methods. Moreover, unlike DPS that applies a likelihood update for every function evaluation. Our method does so only every multiple function evaluations.
>
> > **W3.1**
>
> As we showed in Appendix C.3 "Annealing schedule for $\rho$" paragraph, our annealing schedule decreases to a fixed minimum level at $\rho_{\min}$, so we indeed fixed $\rho$ at small values after certain numbers of iterations. Furthermore, we did run our algorithms for the synthetic prior experiments with a fixed schedule of $\rho$ throughout the process, as shown in Table 4. Experimental results show that our method can accurately sample the posterior, which corroborates our theory. For experiments on linear inverse problems and coded diffraction patterns with FFHQ images, the results with fixed $\rho$ are on par with those with annealing, probably because the problems are not highly non-convex. For more challenging nonlinear problems, such as the Fourier phase retrieval, we empirically find that an annealing schedule is essential to overcome the non-convexity of the problem and provide accurate reconstruction.
>
> > **W3.2**
>
> By combining the above argument with those in the manuscript, we will get the following bound with score function error: $$\frac{1}{T} \int_0^{T} \mathsf{FI}\left(\pi_\tau || \nu_\tau\right) \mathrm{d} \tau \leq \frac{4\mathsf{KL}(\pi^X||\nu_0^X)}{K(1+t^\ast) \min(\rho, \delta)^2} + \frac{1}{K(1+t^\ast)\delta^2}\sum_{k=1}^K\epsilon_{\text{score}}(k).$$

---

> > ### Comment · Reviewer_u5tS · 2024-08-13
> > **Response to authors**
> >
> > I thank the authors for their careful answer.
> > I think that the manuscript would benefit from a more detailed comparison with Ceurdoux et al. and from an extended theoretical analysis with approximate score. Also if should be made clear that the theory is only with fixed rho parameter.   With these clarifications I am ready to recommend acceptance.

---

> > > ### Author Response · Authors · 2024-08-13
> > >
> > > We appreciate your constructive feedback on our work. We will include a more detailed comparison with Coeurdoux et al. and the theoretical analysis with an approximate score function in the final version. We will also clarify that the current theory is only with fixed $\rho$.

---

### Author Rebuttal · Authors · 2024-08-07

## **Response to all reviewers**
We thank all the reviewers for their careful reviews and constructive feedback. We are glad that our method was recognized as "rigorous", "well-executed" (reviewer G2Ty), and “unlike some previous works...effective for both linear and non-linear inverse problems” (reviewer G64m), including “an application to a real imaging problem” (reviewer 43sJ). We are also encouraged that the reviewers found our paper "well-written" (reviewer u5tS) and the writing "extremely clear" (reviewer TmFe).

In the responses below, we address the reviewers' comments individually. Additional experiments are presented in the attached PDF file.

--------------------------------------------------------------------------------

### **Common response to computational efficiency**

We present a comparison of computational efficiency with the major baselines on a linear super-resolution and a nonlinear coded diffraction patterns problem in Response Table 1 below. The clock time in seconds and number of function evaluations (NFE) are calculated for each method to measure its computational efficiency. All hyperparameters are kept the same for each method as those used for Table 1 and Table 2 in the manuscript.

As expected, DM-based approaches (DDRM & DPS) generally yield shorter runtimes due to their lower NFEs. Nevertheless, our PnP-DM method significantly outperforms these methods while achieving comparable runtimes with DPS ($\approx 1.5\times$), despite its larger NFEs ($\approx 3\times$). This is primarily due to two factors: 1) PnP-DM avoids running the full diffusion process by adapting the starting noise level to $\rho_k$ at each iteration, and 2) the runtime is further reduced by using an annealing schedule of $\rho_k$.

We also note that the runtime reported for DDRM and DPS below is the time it takes to generate one sample. For the linear inverse problem experiments, where we generated 20 samples for each sampling method, PnP-DM was faster than DPS because we took 20 samples that PnP-DM generated along one Markov chain of batch size 1 (hence same runtime as below, around 50 seconds) but DPS requires running a diffusion process with batch size 20, which was significantly slower (around 330 seconds).

**Response Table 1**: Comparison of runtime in seconds and number of function evaluations (NFE) for our method and baselines.

|                            | Metric         | DDRM | DPS | PnP-SGS | DPnP  | PnP-DM (ours) |
|----------------------------|----------------|------|-----|---------|-------|---------------|
| Super-resolution           | Clock time (s) | 0.4             | 39             | 20      | 322   | 55           |
|                            | NFE            | 20              | 1000           | 1030             | 18372 | 3032         |
| Coded diffraction patterns | Clock time (s) | --              | 37             | 54      | 261   | 50           |
|                            | NFE            | --              | 1000           | 2572    | 14596 | 2482         |

--------------------------------------------------------------------------------

### **References for all responses**

[1] Coeurdoux et al., 2023. Plug-and-play split Gibbs sampler: embedding deep generative priors in Bayesian inference.

[2] Xu and Chi, 2024. Provably robust score-based diffusion posterior sampling for plug-and-play image reconstruction.

[3] Karras, et al., 2022. Elucidating the design space of diffusion-based generative models.

[4] Chen et al., 2023. Sampling is as easy as learning the score: theory for diffusion models with minimal data assumptions.

[5] Yuan et al., 2023. On a class of Gibbs sampling over networks.

[6] Zhang et al., 2022. Plug-and-play image restoration with deep denoiser prior.

[7] Meng, C., He, Y., Song, Y., Song, J., Wu, J., Zhu, J.Y. and Ermon, S., 2021. SDEdit: Guided image synthesis and editing with stochastic differential equations.

[8] Balasubramanian et al., 2022. Towards a theory of non-log-concave sampling: first-order stationarity guarantees for Langevin Monte Carlo.

[9] Sun et al., 2023. Provable probabilistic imaging using score-based generative priors.

---

### Decision · Program_Chairs · 2024-09-25

**Decision:**

Accept (poster)

**Comment:**

This paper received 5 reviews with balanced views but overall positive. The discussion period has enabled to reach a consensus for acceptance.

All reviewers recognized the quality of the work :
- "the paper is well-written, provides clear and concise explanations of the methods and contributions"
- "The strength of the paper is mainly its experimental section, which covers a range of both linear and non-linear inverse problems."
- "Unlike some previous works, the proposed method is effective for both linear and non-linear inverse problems."
- "The paper provides sufficient theoretical analysis to support the proposed method."
- "The theoretical insights provided, including the stationarity guarantee in terms of average Fisher information, add depth to the method."
- "Experiments show the efficacy of the proposed method compared to existing methods"
- "the experiments are very good; the authors first consider a toy example in which the posterior has a closed form. This serves as a nice introduction to the experimental section ..."
- "this is an original way of solving the diffusion posterior sampling problem."

The authors/reviewers discussion period was useful for resolving the reviewers concerns. Yet, it is required for the final version of the paper that the authors :
- provide a more detailed comparison with Ceurdoux et. al. and clearly state the similarities/differences,
- provide an extended theoretical analysis with approximate score (at least in supplementary material).